# hnRNP H/F drive RNA G-quadruplex-mediated translation linked to genomic instability and therapy resistance in glioblastoma

Pauline Herviou[1,2,3,9], Morgane Le Bras[1,2,3,9], Leïla Dumas[1,2,3], Corinne Hieblot[1,2,3], Julia Gilhodes[4], Gianluca Cioci[5], Jean-Philippe Hugnot[6], Alfred Ameadan [7], François Guillonneau [7], Erik Dassi [8✉], Anne Cammas [1,2,3✉] & Stefania Millevoi [1,2,3✉]

RNA G-quadruplexes (RG4s) are four-stranded structures known to control mRNA translation of cancer relevant genes. RG4 formation is pervasive in vitro but not in cellulo, indicating the existence of poorly characterized molecular machinery that remodels RG4s and maintains them unfolded. Here, we performed a quantitative proteomic screen to identify cytosolic proteins that interact with a canonical RG4 in its folded and unfolded conformation. Our results identified hnRNP H/F as important components of the cytoplasmic machinery modulating the structural integrity of RG4s, revealed their function in RG4-mediated translation and uncovered the underlying molecular mechanism impacting the cellular stress response linked to the outcome of glioblastoma.

[1] Cancer Research Center of Toulouse (CRCT), INSERM UMR 1037, 31037 Toulouse, France. [2] Université Toulouse III Paul Sabatier, 31330 Toulouse, France. [3] Laboratoire d'Excellence "TOUCAN", Toulouse, France. [4] Institut Universitaire du Cancer de Toulouse-Oncopole, 31100 Toulouse, France. [5] TBI, Université de Toulouse, CNRS, INRA, INSA, Toulouse, France. [6] INSERM U1051, Institute for Neurosciences, Hôpital Saint Eloi, Université de Montpellier 2, 34090 Montpellier, France. [7] Plateforme Protéomique 3P5, Université de Paris, Inserm U1016-institut Cochin, Labex GReX, 22 rue Méchain, 75014 Paris, France. [8] Department of Cellular, Computational and Integrative Biology (CIBIO), University of Trento Via Sommarive 9, 38123 Trento, Italy. [9] These authors contributed equally: Pauline Herviou, Morgane Le Bras. ✉email: erik.dassi@unitn.it; anne.cammas@inserm.fr; stefania.millevoi@inserm.fr

Most steps in the gene expression pathway are regulated by the ability of specific RNA regions to form duplexes and other types of RNA conformations involving both Watson-Crick and non-canonical interactions. RNA-binding proteins (RBPs) establish highly dynamic interactions with such RNA elements, creating functional ribonucleoprotein complexes that are essential for every step of post-transcriptional control, including mRNA processing, stability, transport and translation. Accumulating evidences showed that RBPs are dysregulated in cancers, with a major proportion altered in glioblastomas (GBM)[1], one of the deadliest forms of brain cancer, and impact on the expression and function of oncogenic and tumor-suppressor proteins[2]. A detailed knowledge of the interactions between RBPs and their cancer-related RNA targets is vital to better understand tumor biology and potentially unveil new targets for anti-cancer therapy.

Among the many unusual RNA conformations, RNA G-quadruplex (RG4) structures are intriguing not only because they possess unique properties and have been implicated in key cellular functions and gene expression mechanisms but also, and more importantly, their dysregulation has been proposed to have a tremendous impact on human diseases, including cancer[3]. RG4s are extremely stable structures formed by stacking of two or more G-quartets, each composed of four guanines interacting via Hoogsteen bonding. RG4 motif hotspots include both 5′ and 3′ untranslated regions (5′UTR and 3′UTR, respectively)[4], suggesting an important role in mRNA translation. Their formation is regulated by intrinsic properties (e.g., the nature of the coordinating ion, the loop sequence and length, the number of G-quartets) and extrinsic interacting factors, with RBPs being critical regulators of RG4 conformation and function in cancer cells. This notion is supported by studies demonstrating the role of RG4-protein interactions on the expression of cancer-relevant genes[3] as well as by affinity proteomic approaches identifying RG4-binding proteins (or RG4-BPs)[5–9] known to modulate multiple cancer traits. Important insights on the impact of RBPs on RG4 formation have been recently provided by high-throughput RG4 mapping studies which showed that RG4 formation is pervasive in vitro[10,11] but not in cellulo[10]. This led to propose that RBPs might be critical to maintain RG4s unfolded in eukaryotic cells[10]. However, the notion of global in cellulo unfolding is in contrast with cellular imaging studies showing RG4 formation in cellulo as well as with functional in cellulo analysis of RG4-driven endogenous or reporter gene expression[3]. This view has been recently revisited by in vitro transcription experiments[12] and in cellulo RG4s capturing approaches[13] which provided evidence of transient RG4 formation. These observations, together with other findings suggesting that the rate of protein-RNA complex assembly is faster than RG4 structuration[14], reinforced the view that RBPs play a major role in shifting RG4s toward an unfolded state, yet the RG4s dynamics and function remain poorly investigated. Recent unbiased affinity proteomic approaches identified several RG4 interactors, including RBPs (e.g., hnRNP H, hnRNP F, FMRP) and RNA helicases (e.g., DDX21, DDX3X, DHX36)[5–8]. However, given that the strategy used in these studies consisted in comparing RBP binding either to folded G4s or to their mutated version (harboring substitutions of the Gs), the question of which RBPs bind the unfolded RG4s and of whether, how and by what extent they impact on post-transcriptional gene expression in cancer cells have not been fully addressed. Answering these questions is essential to gain a better understanding of the role of RBP-RG4 interactions in translational control where RG4s function as strong repressors by different poorly elucidated mechanisms[15].

Here, we identify hnRNP H and hnRNP F as important components of the cytoplasmic molecular machinery that specifically bind RG4s in their unfolded state. Our findings establish a role for hnRNP H/F as translational regulators acting in synergy with the RNA helicase DHX36 and impacting the biology of GBM. This activity appears to be involved in the resistance mechanisms of GBM, possibly accounting for the failure of current treatments.

## Results

**Identification of the protein machinery binding to folded or unfolded RG4s.** Previous work demonstrated that the canonical RG4 sequence G3A2G3A2G3A2G3 (hereafter referred to as the G3A2) is highly prone to form a RG4 structure in vitro[10,11] but remained largely unfolded when ectopically expressed in cells[10]. This led to propose that RNA helicases and RBPs unfold RG4s and maintain them in an unfolded state. To identify the protein machinery that recognizes RG4 forming G-rich sequences and modulates their function in mRNA translation, we used an unbiased proteomic approach based on RNA affinity purification of cytoplasmic proteins (refer to Supplementary Fig. 1a for fractionation control) with immobilized biotinylated RNAs followed by mass spectrometry (RP-MS). Unlike other studies using RP-MS to identify proteins bound to wild-type RG4-forming or mutated G-less sequences[5–8], we compared affinity enrichment between the G3A2 RNA (G3A2 WT) folded into a RG4 (as described in the Methods section) and its modified version (hereafter referred to as G3A2 7dG) in which replacement of guanines by 7-deaza-guanines prevented Hoogsteen base-pairing and RG4 formation (Supplementary Fig. 1b), as revealed by circular dichroism spectra (Supplementary Fig. 1c). Gel electrophoresis followed by silver staining displayed different complex protein patterns between the native and 7-deaza modified G3A2 RNAs, whereas mock pull-downs with control beads were remarkably clean (Supplementary Fig. 1d). Proteins bound to the G3A2 WT and 7dG RNAs were subjected to tryptic digestion followed by HCD-MS/MS allowing quantitative label free proteomic analysis of RNA-protein interaction data[16]. RG4-BPs (i.e., proteins binding to the G3A2 WT) and G-rich-BPs (i.e., proteins binding to the G3A2 7dG) were defined by the ratio WT/7dG and high confidence proteins (false discovery rate (FDR < 0.05)) were ranked according to an arbitrary 1.5-fold enrichment cutoff after subtraction of the background proteins resulting from non-specific protein binding to the bait RNA sequences (Supplementary Data 1). This quantitative analysis performed with four biological replicates revealed 370 significant G3A2 protein interactors (with 237 proteins found in all replicates), among which we experimentally characterized 328 RG4-BPs and 42 G-rich-BPs. The RP-MS screen (Fig. 1a, Supplementary Data 1) selectively enriched known RG4-BPs, revealed RBPs that have not previously been reported to interact with RG4s and, more importantly, underscored the RBPs that preferentially bind folded or unfolded RG4s. As expected, RNA helicases were found preferentially associated to structured RG4s (Fig. 1a). We compared these results with a recent qualitative RP-MS data set[6] identifying cytoplasmic proteins associated to the RG4 inhibiting NRAS mRNA translation[17]. Of the 370 high-confidence proteins identified in our screen, 27 overlapped with the 80 high-confidence proteins bound to the NRAS RG4[6], resulting in 343 additional cytoplasmic RG4 binders, of which 320 were assigned to specific functional pathways, including translation and RNA metabolism (Fig. 1b, c). In addition, the intersection of our RP-MS and the RNA-binding total proteome (using a compilation of recent RNA interactome capture methods[18–22]), revealed that 260 out of the 370 identified proteins were annotated as RBPs (Fig. 1b and Supplementary Data 2). It is noteworthy that several recently identified m6A-responsive RBPs (based on ref. [20]) were found

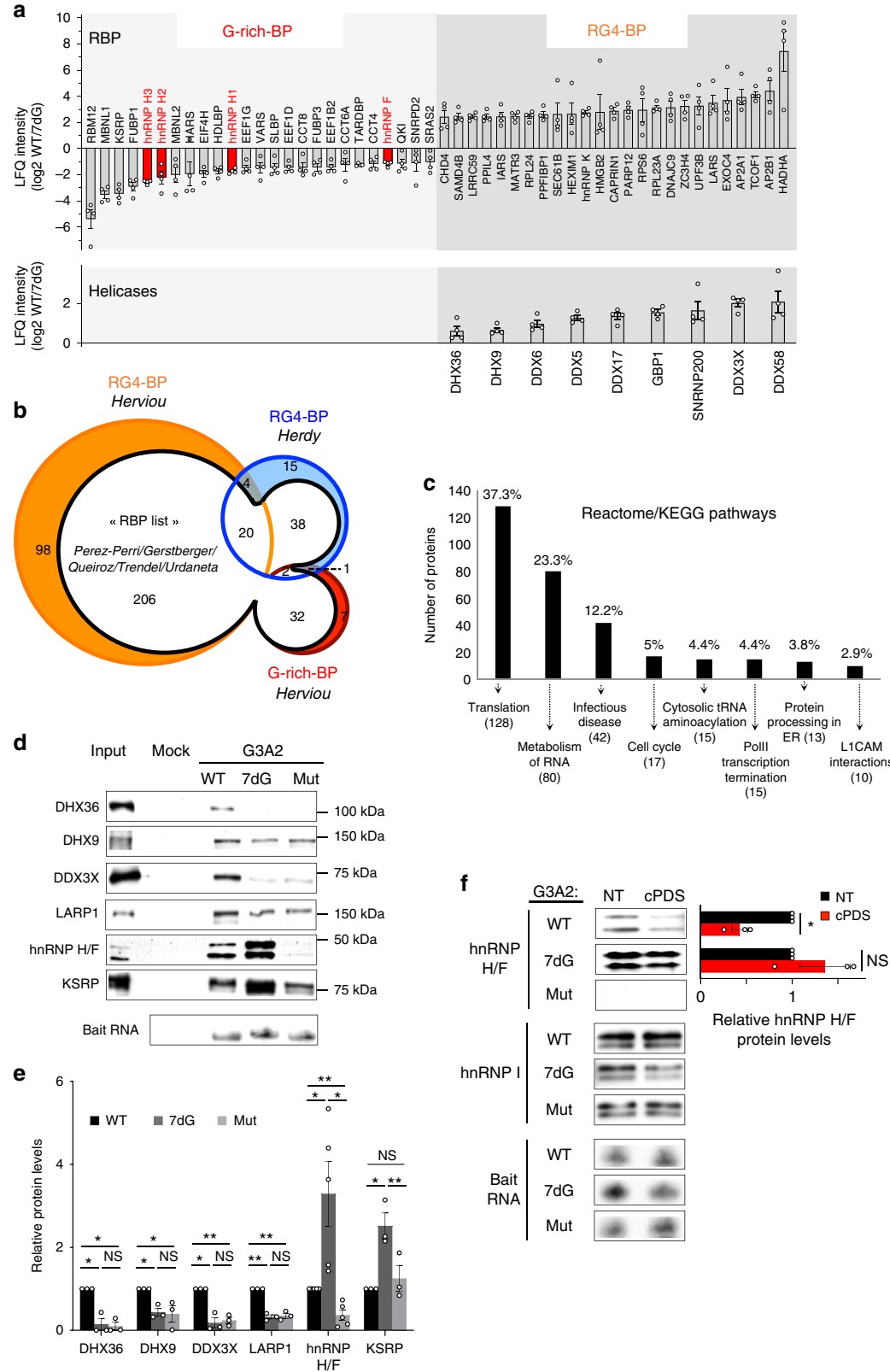

among the RG4-binders, highlighting a possible interplay between RG4s and epitranscriptomic modifications of RNAs[23].

To confirm the RP-MS results and further validate preferential binding to the RG4 sequence or structure, RNA pull-down experiments were repeated by incubating cytoplasmic extracts with RNAs containing the G3A2 WT, the G3A2 7dG or the mutated (G-tract-less or G-less) G3A2 sequence (Mut) (Supplementary Fig. 1b). The recovered proteins, for which the silver stained SDS-PAGE gel displayed distinct protein profiles (Supplementary Fig. 1d), were then probed for some RBPs and

**Fig. 1 Analysis of RG4 sequence or structure binding preferences by RP-MS reveals hnRNP H/F binding to unfolded RG4s. a** RNA affinity chromatography using the G3A2 sequence either native (WT, which forms RG4s) or 7-deaza-modified (7dG, unable to form RG4s) and U251 cytoplasmic cell extracts, followed by mass spectrometry (RP-MS). Proteins identified from RP-MS were sorted according to the ratio G3A2 WT/7dG (top 20 were shown) and to whether they are RBPs (based on refs. [18–22]) or RNA helicases. Data are presented as mean values ± SEM of $n = 4$ independent experiments, FDR < 0.05 (two-sided paired $t$-test). Highlighted in red are the different members of the hnRNP H/F subfamily. **b** Venn diagram showing the overlap of this study (Herviou, red and orange), which identified proteins bound to RG4s either folded (RG4-BPs) or unfolded (G-rich-BP), with the RG4-BPs identified in Herdy[6] (blue) and the RBPs identified in at least 2 RNA capture methods[18–22] (white). **c** Functional enrichment analysis of the identified high confidence 343 factors not known as RG4 binders. **d**, **e** Validation of RP-MS by performing RNA affinity chromatography using G3A2 WT, 7dG or Mut RNAs, followed by western blot analysis (**d**) quantified and normalized to the WT (**e**). Data are presented as mean values ± SEM of $n = 5$ independent experiments for hnRNP H/F and $n = 3$ independent experiments for the other proteins, *$P < 0.05$, **$P < 0.005$, NS: Non-Significant (two-sided paired $t$-test). Bait RNA: RNAs retained on beads. Source data and exact $P$-values are provided as a Source Data file. **f** RNA affinity chromatography using the G3A2 RNAs as in (**d**), treated with carboxypyridostatin (cPDS) or untreated (NT), followed by western blot analysis, quantification and normalization of the hnRNP H/F protein levels to the control (hnRNP I). Data are presented as mean values ± SEM of $n = 3$ independent experiments, $P$-value $= 0.02276$ and $P$-value $= 0.3228$ for the WT and 7dG RNAs, respectively, NS: Non-Significant (two-sided paired $t$-test). Shown is a representative result from $n = 3$ independent experiments. Source data are provided as a Source Data file.

RNA helicases found to bind preferentially either the native or the 7dG G3A2 RNAs (Fig. 1d, e). We focused on proteins known to bind RG4s, as for instance DHX36[24,25], DHX9[24], DDX3X[6], and additional proteins, as KSRP or LARP1, whose function was not associated to RG4 binding. Similar bead-bound RNA elution profiles suggested that the differential interaction was not related to the amount of loaded RNA (Fig. 1d, f). As expected, RG4 helicases recently identified as translational regulators, specifically DHX36[24], DHX9[24], DDX3X[6], were enriched by pull-down with the G3A2 WT and showed less interaction with the G3A2 7dG or the Mut sequence, corroborating the requirement of a RG4 for RNA binding. Similar results were obtained for the translational regulator LARP1 (Fig. 1d, e), who was also found at the NRAS RG4[6], suggesting that this is an uncharacterized RG4-BP possibly binding high G-content 3′UTR motifs[26]. In contrast, the RBPs hnRNP H/F showed a remarkable selectivity towards the 7dG RNA but a weaker interaction with the G-less RNA (Fig. 1d, e), indicating that these proteins bind G-rich sequences incapable of RG4 folding. As observed for hnRNP H/F, KSRP (top 3 hit in Supplementary Data 1), a RBP previously reported to regulate miRNA biogenesis through binding G-rich motifs[27], exhibited stronger binding to the 7dG RNA (Fig. 1d, e) but whether this factor is involved in the RG4 network will require further validation. Overall, these results extended the number of proteins binding the RG4-forming G-rich sequences and provided the first comprehensive evidence of which proteins bind structured RG4s and which ones prefer to bind the G-rich sequence per se.

To bring further insights into the role and mechanism of action of the machinery preferentially binding unfolded RG4s, we focused on two closely (structurally and functionally) related RBPs[28–31], hnRNP H and hnRNP F (or hnRNP H/F), since these factors have been reported to regulate mRNA expression through binding RG4-forming sequences[29,32,33] but their role in translation via these motifs or structures has not been investigated yet. As observed for the G3A2, hnRNP H/F binding to the NRAS RG4[17] depended on RG4 unfolding and the presence of G-stretches (Supplementary Fig. 2). It is interesting to note that the overall binding protein profile was similar between NRAS and G3A2 but differed between cytoplasmic and total extracts (Supplementary Fig. 2b, c). The RG4 structuration-dependency of hnRNP H/F binding was further analyzed by RNA-pull down with RNA baits pre-incubated with either the small-molecule ligand carboxypyridostatin (cPDS) or pyridostatin (PDS) known to specifically stabilize cytoplasmic RG4s[34] or RNA/DNA G4s[35], respectively. We found that the binding of hnRNP H/F, but not that of the control polypyrimidine tract-binding protein hnRNP I (described in the Methods section), to both the G3A2 RG4 (Fig. 1f and Supplementary Fig. 3a) and NRAS RG4

(Supplementary Fig. 2d) was decreased upon cPDS or PDS treatment. Similar results were obtained by reversed pull-down (i.e., RBP/helicase immunoprecipitation of G3A2 RNAs (WT or 7dG), followed by RNA detection; Supplementary Fig. 3b, c) and surface plasmon resonance (Supplementary Fig. 3d, e), further validating that the binding of hnRNP H/F and helicases depend on RG4 structuration.

**hnRNP H/F localization and association with translationally active fractions.** The ability of cytoplasmic hnRNP H/F to bind to unfolded RG4s prompted us to study the function and mechanism of action of these interactions in regulating mRNA translation in cancer cells. We focused on high-grade glioma or GBM, highly aggressive, angiogenic and treatment-resistant brain tumors, for the following reasons. First, previous studies showed that RBPs are highly dysregulated in GBM[1], with hnRNP H/F being over-expressed both at the protein and mRNA level[28,36]. Then, the nuclear activity of these factors appeared to be involved in the pathogenesis and progression of malignant gliomas[28]. Finally, it is well known that mRNA translation dysregulation contributes to GBM progression and response to current therapeutic treatments[37,38], yet the molecular mechanisms and therapeutic targets remain to be fully elucidated. To address whether hnRNP H/F drive translational control of genes contributing to GBM progression and treatment, we first evaluated the expression level of hnRNP H/F in tumor and normal tissues from the TCGA database. We found that hnRNP H/F family members displayed higher expression levels in GBM compared to normal brain (Supplementary Fig. 4a), suggesting a potential role for both RBPs in GBM gene expression reprogramming. In addition, data from REMBRANDT (Repository for Molecular Brain Neoplasia Data), a publicly available dataset with information on tumor gene expression, treatment history, and survival, demonstrated that high hnRNP H or hnRNP F expression is correlated with poor survival (Supplementary Fig. 4b), indicating that hnRNP H/F are likely clinically relevant molecular target in GBM. To gain insight into the role of hnRNP H/F in translation regulation in GBMs, we first addressed their specific localization by subcellular fractionation of three GBM cell lines (U251, LN18 and U87) that differ in their response to chemo- and radiotherapy treatments and in the mutational profiles (Supplementary Fig. 4c). In addition to being present in nuclear fractions, hnRNP H/F co-distributed with proteins associated with active translation (eIF4A) and was enriched in microsomal fractions, containing endoplasmic reticulum-associated proteins (Fig. 2a). This result is consistent with previous findings showing moderate to high cytoplasmic expression for both hnRNP H and hnRNP F,

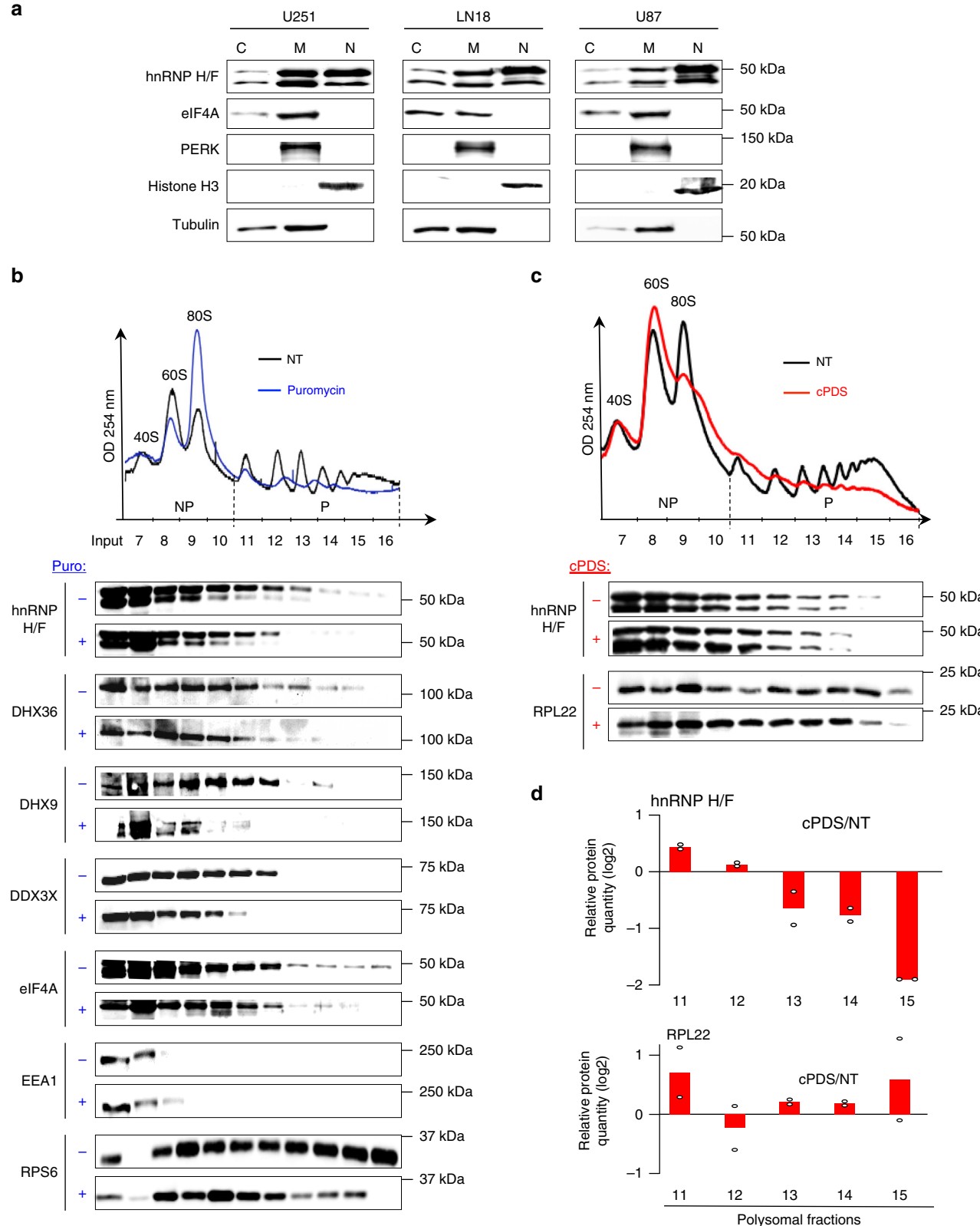

depending on the tissues and on the normal/tumoral status[39]. Then, we performed polysomes profiling combined with immunoblotting to monitor the distribution of hnRNP H/F between translational inactive (non-polysomes, NP) and active (polysomes, P) fractions, in the absence or presence of puromycin, a drug that causes ribosome dissociation. We found that hnRNP H/

F and other RG4 helicases (e.g., DHX36, DHX9 and DDX3X) co-sedimented with translating polyribosomes and that their association depended on polysome integrity (Fig. 2b). The co-sedimentation profiles observed and their modifications after treatment with puromycin were similar to those of the initiation factor eIF4A. Specifically, the fraction of hnRNP H/F loaded on

**Fig. 2 hnRNP H/F subcellular localization and association with polysomal fractions. a** Subcellular fractionation of GBM cell lines, followed by western blot analysis of hnRNP H/F, eIF4A (cytosolic and microsomal marker), PERK (microsomal marker), histone H3 (nuclear marker) and tubulin (cytosolic marker associated to microsomes). Nuclear (N), microsomal (M), and cytosolic fractions (C). Shown is a representative result from $n = 2$ independent experiments. Source data are provided as a Source Data file. **b** Polysome profile of U251 cells untreated (NT) or puromycin treated (Puro), followed by western blot analysis from individual non-polysomal (NP) and polysomal (P) fractions by probing for hnRNP H/F, DHX36, DHX9, DDX3X, eIF4A. EEA1: negative control. RPS6: positive control. Shown is a representative result from $n = 3$ independent experiments. Source data are provided as a Source Data file. **c** As in **b**, except that cells were NT or treated with 20 µM carboxypyridostatin (cPDS) for 1 h, and probing for hnRNP H/F or RPL22 (negative control). Shown is a representative result from $n = 2$ independent experiments. Source data are provided as a Source Data file. **d** Repartition of hnRNP H/F proteins in polysomal fractions was quantified with $n = 2$ independent experiments.

the polysomes was 5% and, similarly to Sauer et al.[25], those of DHX36 and eIF4A were 9.5 and 7%, respectively. These results suggest the involvement of hnRNP H/F in the regulation of the initial steps of mRNA translation.

Consistent with the observation that RG4 structuration reduced hnRNP H/F RNA-binding (Fig. 1), we observed that the cPDS- (or PDS- (Supplementary Fig. 5a)) induced stabilization of RG4s resulted in the dissociation of hnRNP H/F from translating ribosomes in U251 (Fig. 2c, d, Supplementary Fig. 5a) and U87 (Supplementary Fig. 5b) GBM cells. Taken together, these results suggest that hnRNP H/F localize to sites of active translation and associate to translating ribosomes in a way that depends on the ability of RG4s to adopt an unfolded conformation.

**Role for hnRNP H/F in translational regulation of DNA damage response genes.** To demonstrate a functional role for hnRNP H/F in translational regulation, we transfected U87 or LN18 GBM cells with hnRNP H and/or hnRNP F specific or control siRNAs for 48 h, followed by quantification of global protein synthesis rates by pulse-labeling with puromycin and immunoblotting using an anti-puromycin antibody (i.e., SUnSET assay). We found that hnRNP H/F silencing induced only minor reduction of global translation rates (Supplementary Fig. 6a–d). Consistent with this, the polysomal profile was slightly altered by hnRNP H/F depletion (Fig. 3a, Supplementary Fig. 6e), indicating that cells deficient in hnRNP H/F are not globally defective in protein synthesis. Neither apoptosis nor proliferation were affected under these treatment conditions (Supplementary Fig. 6f, g), suggesting that changes in translational efficiency after hnRNP H/F silencing were not directly related to these processes. Based on these results and our previous findings (Figs. 1 and 2), we reasoned that hnRNP H/F might selectively control translation of sub-groups of mRNAs containing RG4-forming sequences. To test this hypothesis, we first mapped RG4-forming sequences within hnRNP H/F-binding regions in 5′UTRs, 3′UTRs and CDSs by combining the bioinformatic prediction of RG4 formation (using QGRS Mapper[40]) and the reanalysis of previously published in cellulo RNA-protein interactions using CLIP-seq (cross-linking immunoprecipitation (CLIP) combined with deep RNA sequencing) data[41,42] (Supplementary Fig. 7a). Strikingly, hnRNP H/F bound an important fraction of RG4s over all the RG4s predicted in the transcriptome (11% of 5′UTR, 2.7% of CDS, and 11.4% of 3′UTR) (Fig. 3b). Similar results were obtained by intersecting experimentally validated RG4s (based on[11]) with hnRNP H/F CLIP-seq data (Supplementary Fig. 7b, c), although the magnitudes of the enrichment were different and reflected the shifted abundance of RG4s identified in the different regions of the mRNA by the rG4-seq method[11] (Supplementary Fig. 7b). Overall, these results support the notion of widespread regulation of RG4-containing mRNAs by hnRNP H/F. In addition, RG4s were significantly enriched in the binding regions of hnRNP H/F relative to random control sequences (Fig. 3c, Supplementary

Fig. 7d and Supplementary Data 3). Most hnRNP H/F sites in those regions contain a high-scoring RG4-forming sequence (Supplementary Fig. 7e), with hnRNP F sites being less dense but still highly enriched, especially in 5′UTRs (Fig. 3c). These results extend the notion of a physical link between hnRNP F and RG4s (recently investigated in ref. [32]) to translational regulatory regions, but most notably underscore the extent of hnRNP H-RG4 interactions, which has not been reported so far. Gene Ontology enrichment analysis showed that hnRNP H and hnRNP F bind RG4-containing RNAs associated with genes regulating cell stress response, including those involved in the response to DNA damage (DDR) (Supplementary Fig. 7f). This result is particularly relevant to GBM since chemo- and radio-resistance of these tumors is largely influenced by the expression of DDR genes[43]. We then asked whether RG4-containing mRNAs bound by hnRNP H/F were candidates for direct translational control by these factors. To this end, we performed polysomal fractionation of hnRNP H/F-depleted cells followed by RNA isolation from non-polysome (NP), light (LP) and heavy (HP) polysome fractions and RT-qPCR analysis. Based on our bioinformatic analysis (Fig. 3b, c and Supplementary Fig. 7), we selected 5 mRNAs involved in the DDR and/or playing a function in GBM that contained an hnRNP H/F binding site overlapping RG4-forming sequences. Among them, the mRNA encoding VEGF (vascular endothelial growth factor) was chosen as positive control due to its pivotal role in regulating tumor angiogenesis in human gliomas[44]. Also, the VEGF mRNA is regulated at the translational level by a variety of mechanisms relying on different sequence/structure elements, including RG4s[45]. Furthermore, we previously demonstrated that RG4 stabilization strongly represses VEGF mRNA translation and protein expression in living cells[45]. The ability of these mRNAs to form RG4s was validated by performing RNA-immunoprecipitation (RIP) assays with cytoplasmic extracts and the BG4 antibody, known to recognize folded RG4s[34]. In agreement with the bioinformatic analysis of RG4 formation, we found that these mRNAs were prone to form RG4s in cellulo (Fig. 3d and Supplementary Fig. 8a). The translational efficiency of these mRNAs and the control HPRT mRNA, following hnRNP H/F silencing, was quantified either by analyzing the ratio HP/total RNA (Fig. 3e) or by measuring the distribution of each mRNA across the gradient (Supplementary Fig. 8b). We observed that hnRNP H/F depletion induced a significant modification in mRNAs association with translating polysomes, indicating a role of hnRNP H/F in both translational activation (MECP2 and PRR5) and repression (VEGF, USP1, CCNA2, BABAM1) (Fig. 3e and Supplementary Fig. 8b). Importantly, cPDS cellular treatments over short periods of time (1 h) also modified the translation efficiency of these targets (Fig. 3f), without affecting the mRNA amounts for all except the USP1 mRNA, for which the effect on transcripts levels was reversed compared to the translational effect (Supplementary Fig. 8c). For this target, and in agreement with previous findings obtained with the VEGF mRNA[45], we further validated the direct effect of cPDS on RG4-dependent translation using USP1 RNA

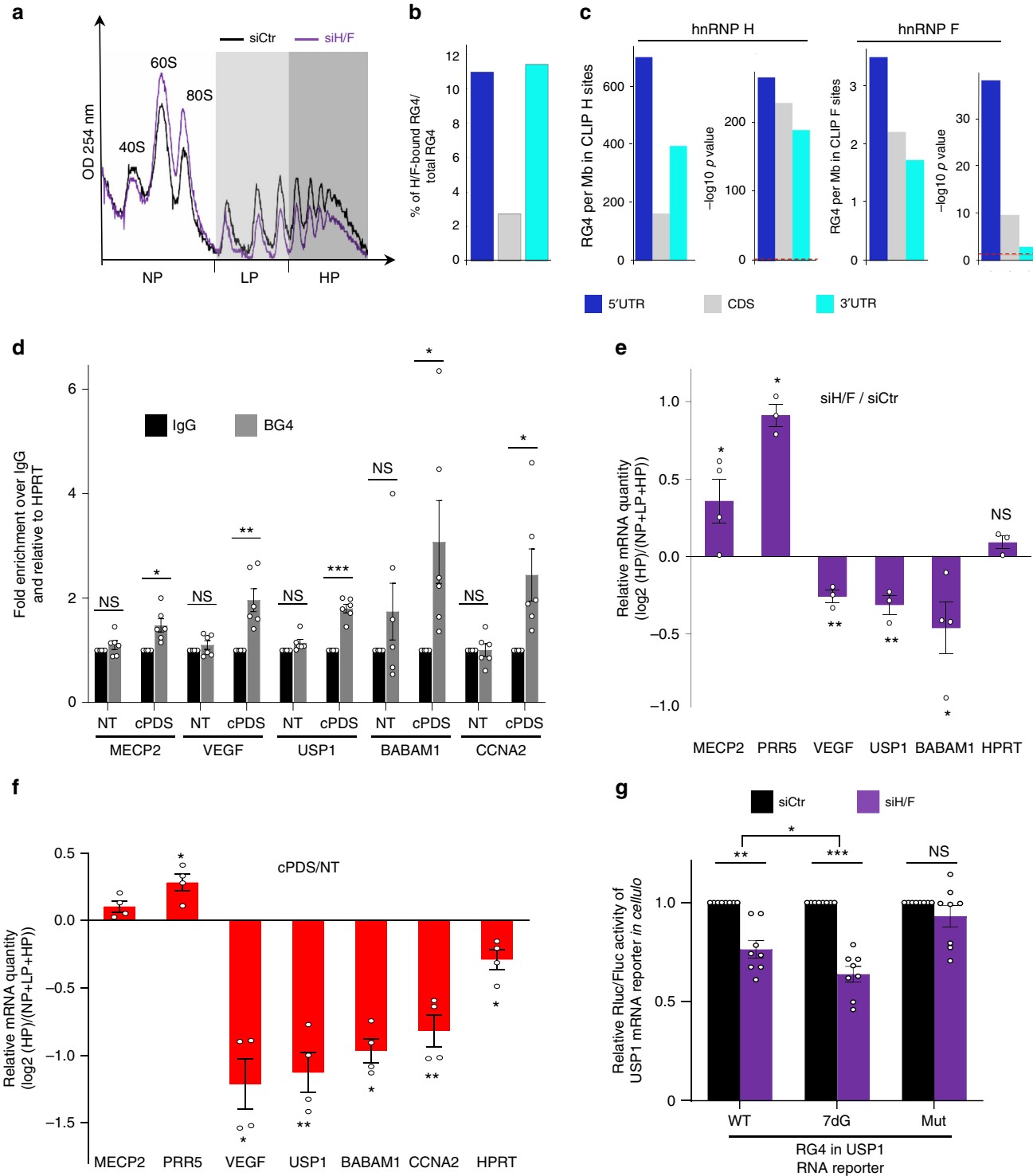

reporters and in vitro translation assays (Supplementary Fig. 8d). It is important to note that hnRNP H/F depletion and cPDS treatment resulted in similar positive/negative effects in translation efficiency (Fig. 3e, f) that were consistent with the ability of cPDS to both diminish hnRNP H/F binding to RG4 forming sequences (Fig. 1 and Supplementary Figs. 1–3) and dissociate hnRNP H/F from translating ribosomes (Fig. 2c, d). To further explore the translational regulation mediated by hnRNP H/F bound to RG4 forming sequences, we transfected GBM U87 ((Fig. 3g) or U251, Supplementary Fig. 8e) cells with in vitro-transcribed USP1 RNA reporters containing the RG4 sequence

WT (USP1 RG4 WT), 7dG-modified (USP1 RG4 7dG) or mutated (USP1 RG4 Mut). We observed that hnRNP H/F silencing significantly decreased the expression of the USP1 RG4 WT, but also, by a greater extent, that of the USP1 RG4 7dG, while leaving the USP1 RG4 Mut reporter unaffected (Fig. 3g, Supplementary Fig. 8e). Moreover, ligand-induced RG4-stabilization resulted in significant inhibition of the USP1 WT, not the USP1 RG4 7dG, expression (Supplementary Fig. 8f). These functional effects fully mirrored both the efficiency of hnRNP H/F binding to the RG4 RNAs (WT, 7dG, Mut) (Fig. 1d, e, Supplementary Figs. 2 and 3) and the effect of a RG4-stabilizing

**Fig. 3 hnRNP H/F drive mRNA translation of stress-response genes. a** Polysome profile of U87 cells treated with control (siCtr) and hnRNP H/F (siH/F) siRNAs. The positions of the 40S, 60S and 80S ribosomal subunits and non-polysomal (NP) and light (LP) and heavy (HP) polysomal fractions are indicated. **b** Fraction of RG4s (in 5′UTR, CDS, and 3′UTR regions) bound by hnRNP H/F over all RG4s predicted in the transcriptome. **c** Density of RG4s per Mb of hnRNP H and F binding sites, along with the -log10(P-value) of the enrichment with respect to random sites. **d** Immunoprecipitation (IP) of in cellulo RNA-protein complexes in U87 cells (cytoplasmic fraction) untreated (NT) or treated with 20 μM carboxypyridostatin (cPDS) for 2 h using the BG4 antibody or control IgG, followed by RT–qPCR analysis. The relative mRNA levels for each IP sample were normalized to the corresponding IP IgG and to the corresponding input sample and were plotted relatively to the HPRT mRNA (negative control). Data are presented as mean values ± SEM of $n = 3$ independent experiments. **e** As in **a**, but followed by RT–qPCR analysis from pooled NP, LP, HP fractions, for the indicated mRNAs and quantification by analyzing the ratio HP/total mRNAs. Data are presented as mean values ± SEM of $n = 3$ independent experiments. **f** NP, LP, HP fractions were extracted from U87 cells NT or treated with 20 μM cPDS for 1 h and RT-qPCR was performed using primers for the indicated mRNAs. Quantification and plot as in **d**. Data are presented as mean values ± SEM of $n = 4$ independent experiments. **g** Ratio of Renilla/Firefly luciferase activities (Rluc/Fluc) determined using U87 cells treated with siCtr and siH/F siRNAs, followed by cotransfection with USP1 RNA reporters containing the RG4 unmodified (WT), 7dG-modified (7dG) or mutated (Mut) and an internal control mRNA encoding the Fluc. Data are presented as mean values ± SEM of $n = 4$ independent experiments. For all the panels, *P < 0.05, **P < 0.005, ***P < 0.0005, NS: Non-Significant (two-sided paired t-test). For **a**, **d**, **e**, **f**, **g** data and exact P-values are provided as a Source Data file.

ligand on these interactions (Fig. 1f, Supplementary Figs. 2 and 3). The observations that the translation of the USP1 mRNA driven by the unfolded RG4 (7dG-modified) was insensitive to stabilizing ligands (Supplementary Fig. 8f) but much more responsive to hnRNP H/F loss compared to the USP1 RG4 WT (Fig. 3g, Supplementary Fig. 8e) suggest that the dynamic equilibrium between RG4s and linear G-rich sequences in cellulo results in low binding of hnRNP H/F to RG4s but, when preventing RG4 from folding, hnRNP H/F strongly bind the G-rich RNA to potentiate translation. In agreement with the dual role of RG4s in translation[15], these results also demonstrate that, RG4 stabilization, resulting from either the absence of hnRNP H/F or from the addition of RG4 stabilizing ligands, can either activate or suppress mRNA translation.

**Cooperation between hnRNP H/F and DHX36 in translational regulation.** We then sought to define the molecular mechanism underlying the function of hnRNP H/F in translation regulation involving RG4 motifs. Previous work speculated on the possibility that hnRNP H/F-RG4 interaction could be facilitated by helicases, specifically DHX36[46]. This unwinding factor has been shown to bind RG4s both in vitro[47] and in cellulo[48], and to be required for optimal translation of two mixed lineage leukemia proto-oncogenes in synergy with Aven[49]. Furthermore, DHX36 is associated with translating polysomes (Fig. 2b) and regulates mRNA translation by specifically targeting RG4s[24]. To investigate the possibility that hnRNP H/F and DHX36 cooperate to regulate RG4-dependent translation, we first performed co-immunoprecipitation assays using total (TE) or cytoplasmic (CE) extracts from U87 (Fig. 4a) or U251 (Supplementary Fig. 9a) GBM cells, in the presence of RNase and DNase to exclude nucleic acid-mediated interactions. In agreement with previous large-scale protein-protein interaction studies[50], we found that hnRNP H/F co-immunoprecipitated with DHX36 in both total and cytoplasmic extracts, irrespective of which protein was immunoprecipitated (Fig. 4a and Supplementary Fig. 9a). Unlike DHX36, DHX9 and DDX3X were co-immunoprecipitated with hnRNP H/F in total extract, but weakly in the cytoplasmic extract, suggesting the formation of different RBP-helicase-RG4 complexes depending on their subcellular localization. However, neither hnRNP H/F nor DHX36 antibodies precipitated the translation initiation factor eIF4A (Fig. 4a and Supplementary Fig. 9a), recently proposed as an RG4 regulator[51]. To analyze the formation of ribonucleoprotein complexes involving hnRNP H/F, DHX36 and RG4-containing mRNAs, we performed a series of RIP assays using cytoplasmic extracts from U87 cells. In agreement with CLIP-data[41,42], we found that the hnRNP H/F antibody immunoprecipitated endogenous mRNAs (Fig. 4b)

previously identified as hnRNP H/F translational targets (Fig. 3). Since these mRNAs were also found in DHX36 RIP samples (Fig. 4b), we concluded that hnRNP H/F-DHX36 interactions might be involved in the translation regulation of RG4-containing mRNAs. However, as shown above, even if the two proteins shared similar distribution profiles in polysomes (Fig. 2b) and in microsomes (Fig. 2a and Supplementary Fig. 9b), they display opposite RNA-binding preferences, with hnRNP H/F preferentially associated to unfolded RG4s while DHX36 showing an improved association to structured RG4s (Fig. 1 and Supplementary Data 1). By combining RIP with the depletion of either of these factors, we tested the possibility of a sequential mechanism that would first unfold the RG4s and then keep them unfolded. As shown in Fig. 4c, d and Supplementary Fig. 9c, d, while DHX36 silencing reduced the binding of hnRNP H/F to RG4 targets, the recruitment of DHX36 was not affected by hnRNP H/F depletion, indicating that DHX36 is necessary for hnRNP H/F to bind to RG4s targets but not the opposite. Together, these results suggest that hnRNP H/F is recruited onto G-rich elements through direct interaction with DHX36 once the latter has bound and unfolded RG4s. To further test this model, we verified the in cellulo RG4 structuration after depletion of hnRNP H/F or DHX36 in LN18 (or U251), using the BG4 antibody and the treatment with cPDS as a positive control[34]. For both cell lines, we observed that depletion of either of the two factors induced a similar increase in the BG4 signal, which was RNAse-dependent and comparable in magnitude to that previously observed for DHX36[25] (Fig. 4e, f and Supplementary Fig. 9e). Therefore, hnRNP H/F and DHX36 might cooperate to maintain RG4s in an unfolded conformation, thus facilitating or repressing mRNA translation depending on whether the specific RG4 plays a negative or a positive role in this process, respectively.

**Impact of hnRNP H/F-RG4 mediated translational regulation on the DDR.** Based on the observation that a sub-group of mRNAs containing RG4 and interacting with hnRNP H/F are associated with stress response (Supplementary Fig. 7f), we hypothesized that the RG4 formation induced by hnRNP H/F silencing or RG4 stabilization (Fig. 4e, f and Supplementary Fig. 9e) could interfere with the cells' ability to synthesize proteins playing a cytoprotective role, resulting in cellular DNA damage stress. Combined analysis of two markers of genetic instability, γ-H2AX (i.e., phosphorylated H2AX) and 53BP1[52], by immunofluorescence microscopy revealed that hnRNP H/F removal from LN18 cells induced the appearance of nuclear foci of both factors (Fig. 5a). Consistent with this result, increased phosphorylation of H2AX was observed after treatment of LN18 cells with cPDS

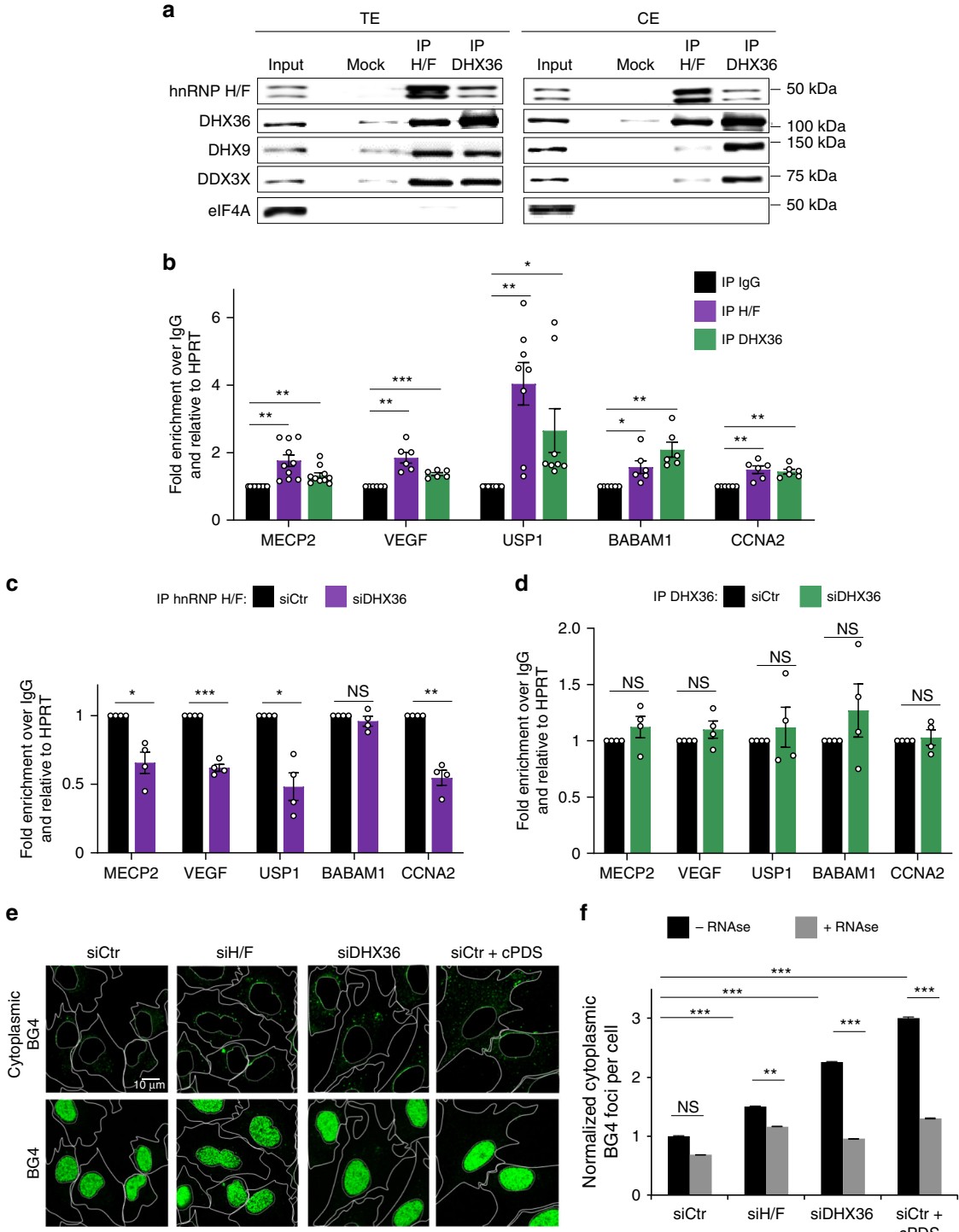

(Fig. 5b, c). Ionizing radiation radiotherapy (IR) and temozolo-mide chemotherapy (TMZ) are part of the standard treatment for GBM patients. The ability of hnRNP H/F to induce DDR markers prompted us to determine whether hnRNP H/F silencing could enhance the cytotoxic effects of IR or TMZ. To this end, we depleted hnRNP H/F in the chemo- and radio-resistant GBM cell line LN18 and either analyzed DNA damage repair by monitoring H2AX phosphorylation or measured the cell survival fraction using clonogenic assays, after treatment with IR (Fig. 5d, e and Supplementary 10a–c) or TMZ (Supplementary 10d, e). Our results showed that hnRNP H and/or hnRNP F silencing induced a marked increase in γ-H2AX after exposure to IR (Fig. 5d and

Supplementary Fig. 10a, b). Similar results were observed in the presence of the chemotherapeutic agent TMZ (Supplementary Fig. 10d). This reduced ability to cope with genotoxic stress correlated with reduced survival fraction upon IR (Fig. 5e and Supplementary Fig. 10c) or TMZ (Supplementary Fig. 10e). To define whether the LN18 cells' ability to recover after genotoxic stress is dependent on translational regulation by hnRNP H/F, we analyzed the expression of γ-H2AX after treatment with TMZ, in the presence or absence of ectopically expressed hnRNP H/F and/ or the translational inhibitor cycloheximide (CHX). As indicated in Fig. 5f, we observed that γ-H2AX was increased after CHX treatment, indicating that the recovery from TMZ-induced DNA

**Fig. 4 hnRNP H/F collaborate with DHX36 to regulate RG4-dependent translation. a** Immunoprecipitation (IP) of U87 total (TE) or cytoplasmic (CE) extracts, followed by western blot analysis and probing with the indicated antibodies. Shown is a representative result from $n = 3$ independent experiments. Source data are provided as a Source Data file. **b** IP of in cellulo RNA-protein complexes (RIP) in cytoplasmic extracts from U87 cells with the hnRNP H/F or DHX36 antibody, followed by RT-qPCR analysis of MECP2, VEGF, USP1, BABAM1, CCNA2, HPRT mRNAs. Data are presented as mean values ± SEM of $n = 5$ independent experiments for MECP2 and $n = 3$ independent experiments for the other mRNAs, *$P < 0.05$, **$P < 0.005$, ***$P < 0.0005$, (two-sided paired $t$-test). **c**, **d** RIP as in **b** but after treatment with control (siCtr) siRNAs and either DHX36 (siDHX36) (**c**) or hnRNP H/F (siH/F) (**d**) siRNAs, followed by RT-qPCR analysis. The relative mRNA levels for each RIP sample in (**b**–**d**) were normalized to the corresponding IP IgG and input sample, and were plotted relatively to the HPRT mRNA. Data are presented as mean values ± SEM of $n = 4$ independent experiments, *$P < 0.05$, **$P < 0.005$, ***$P < 0.0005$, NS: Non-Significant (two-sided paired $t$-test). **e** Immunofluorescence experiments in LN18 cells using the BG4 antibody after treatment with siCtr, siH/F, siDHX36 siRNAs and carboxypyridostatin (cPDS). Phase contrast served to mark the cytoplasm and the nucleus. Panels with masked nuclear signal allow visualization of the BG4 signal in the cytoplasm. Shown is a single representative field from one experiment over $n = 2$ independent experiments. **f** Quantification of BG4 cytoplasmic foci number per cell observed in **e**. Number of cells counted in the -RNase conditions: 7132 cells for siCtr, 4945 cells for siH/F, 7877 cells for siDHX36, 6843 cells for siCtr+cPDS; Number of cells counted in the +RNase conditions: 6844 cells for siCtr, 5901 cells for siH/F, 6770 cells for siDHX36, 6893 cells for siCtr+cPDS. Data are presented as mean values ± SEM, statistical significance was performed on the full cell populations *$P < 0.05$, **$P < 0.005$, ***$P < 0.0005$, NS: Non-Significant (two-sided Kolmogorov–Smirnov test). For **b**–**d**, **f** source data and exact $P$-values are provided as a Source Data file.

damage was dependent on protein synthesis. Overexpression of hnRNP H/F markedly reduced H2AX phosphorylation, suggesting that these factors play a role in the recovery after TMZ-induced DNA damage. The observation that this effect is counteracted by CHX, led us to propose that hnRNP H/F control the cell response to a genotoxic insult by regulating the synthesis of proteins involved in the DDR.

**USP1 translational regulation by hnRNP H/F and DHX36 in GBM cells and tumors**. To strengthen the notion that hnRNP H/F and DHX36 cooperate to regulate the translation of RG4-containing DDR genes, we decided to focus on the mRNA encoding USP1, an ubiquitin peptidase with important functions in DNA repair[53,54]. USP1 was chosen for further investigation also because its increased expression in GBM has been associated to resistance to treatments[55], providing a rationale for USP1 inhibition as a potential therapeutic approach against GBM. Furthermore, the USP1 mRNA translational regulation has been recently identified as a major mechanism of cisplatin resistance in non-small-cell lung cancer, yet the molecular mechanisms remain to be investigated[56]. We first validated that, as shown for hnRNP H/F (Fig. 3), USP1 is a DHX36 translational target by performing polysomal analysis combined RT–qPCR analysis of the USP1 mRNA. We found that the polysome profile of U87 was only slightly altered by DHX36 depletion (Fig. 6a), in agreement with previous findings reporting a mRNA specific role of this helicase in translational regulation[24]. In agreement with polysomal analysis of USP1 mRNA translation regulation by hnRNP H/F (Fig. 3e and Supplementary Fig. 8b) or DHX36 (Fig. 6b and Supplementary Fig. 11a), silencing of either of these factors or treatment with cPDS reduced USP1 protein expression (Fig. 6c–e and Supplementary Fig. 11b), providing further support for a RG4-dependent translational mechanism in which both hnRNP H/F and DHX36 cooperate to activate USP1 protein synthesis. In addition, loss of DHX36 or hnRNP H/F induced an increase in protein ubiquitination, in agreement with USP1 deubiquitinating function (Fig. 6c).

Finally, to investigate the potential clinical importance of our findings, we analyzed the expression of hnRNP H/F, DHX36 and USP1 in human glioma patient tissues. Gliomas are classified into low-grade (LGG) types with slow growth, and high-grade types (HGG) (or GBM), with fast growth and spread into normal brain tissue[57]. Analysis of the protein expression of the three factors in four LGG and three GBM human tumor samples revealed that hnRNP H/F, DHX36 and USP1 were markedly more expressed in GBM compared to LGG. In HGG, the fluctuation in the protein expression of USP1 appeared to correspond to that of hnRNP H/

F and DHX36 (Fig. 6f). These results, together with the observation that hnRNP H/F and DHX36 interacted in the cytoplasm (Fig. 4a, Supplementary Fig. 9a), that both factors bound the USP1 mRNA and controlled its protein expression (Figs. 4b–d and 6b–d and Supplementary Fig. 11), strongly support a role for hnRNP H/F and DHX36 in coordinating USP1 expression in GBM.

## Discussion

Recent data proposed that RG4s tend to massively form in vitro[10,11], in accordance with their great stability, but their in cellulo formation was proposed to be highly dynamic due to the presence of a protein machinery that drive them to an unfolded state[10,13]. In contrast to previous RP-MS data sets[5–9], we were able to capture and identify proteins binding to folded and unfolded RG4s by comparing native and 7dG-substituted G3A2 RNAs. Of note, incorporation of 7dG was instrumental in the identification of functionally relevant G4s in long RNAs[58].

Our RP-MS screen (Supplementary Data 1) selectively enriched several RNA helicases (e.g., DHX36, DHX9, DDX3X, DDX5, DDX17) (Fig. 1), reinforcing the concept of a dynamic equilibrium between the formation and resolution of RG4 structures. Surprisingly, while eIF4A, who was previously found to be required for translation of two-quartet RG4-forming $(CGG)_4$ motifs[51], did not associate with RG4s (as in ref. [6]), its co-factor, eIF4H, selectively bound the 7dG G3A2 RNA. In agreement with[59], eIF4H could help to destabilize the RG4 by binding to the newly formed single-stranded region after partial strand structure unfolding by eIF4A. However, the observation that hnRNP H/F did not interact with eIF4A (Fig. 4a and Supplementary Fig. 9a) and bind G triplets[60] susceptible to structure in three-quartet RG4s, suggests the intriguing possibility that the requirement of a specific helicase-RBP pairs (eIF4A-eIF4H or DHX36-hnRNP H/F) depends on the number of quartets stacked to form RG4s. The intersection of our RP-MS data with the NRAS RG4-binding cytoplasmic proteome[6] and the RNA-binding total proteome[18–22] (Supplementary Data 2) revealed cytoplasmic RBPs whose function was not associated to RG4-binding, including known translation factors, such as LARP1[26]. We also identified additional RG4-binding proteins, including known RNA-interactors but also proteins that have not been annotated as RBPs, thus extending the number of proteins binding the RG4-forming G-rich sequences (Fig. 1, Supplementary Data 1 and 2). Future studies will be needed to fully characterize the RG4/G-rich binding proteome in terms of specificity, selectivity, RG4/G-rich topology and mode of binding (direct or indirect).

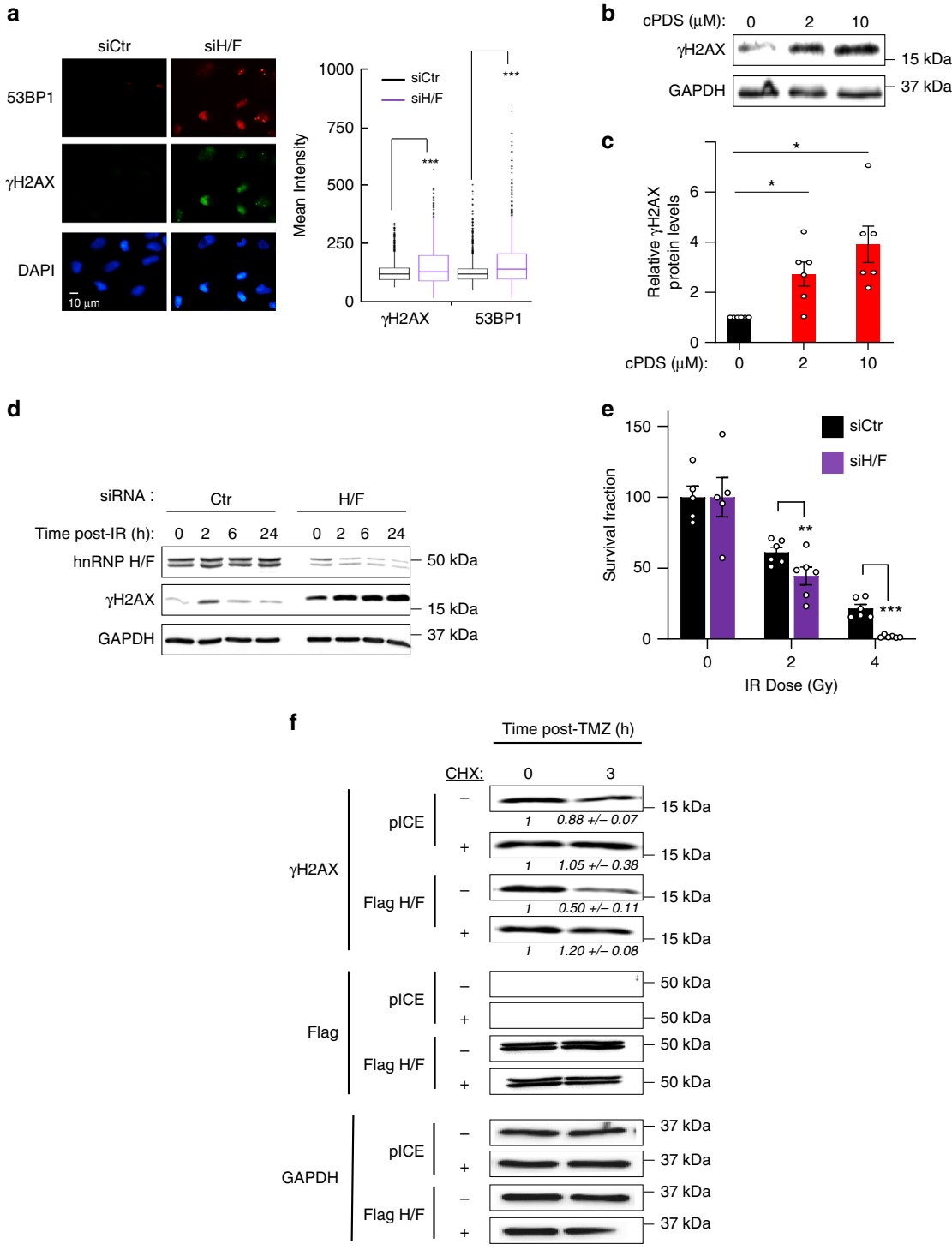

More importantly, our work underscored the RBPs that preferentially bind unfolded RG4s, which included all hnRNP H/F family members (Supplementary Data 1). This result is consistent with RNA-protein interaction studies using either purified recombinant proteins[14,61] or extracts from GBM (U87) cells[33] showing that hnRNP H[33] and hnRNP F[14,61] preferentially binds linear G-tracts. The *Drosophila* hnRNP H/F homolog, Glorund, also recognizes G-tracts RNA in a single-stranded conformation[62]. In contrast, other sets of studies demonstrated that hnRNP H and/or hnRNP F[29,32] bind RG4s, but not the mutated version, and that the small molecule TMPYP4 modulates this

interaction[29,32]. To reconcile this whole set of results, and based on the observation that hnRNP H/F binding is modulated by DHX36 silencing but not the opposite (Fig. 4), we propose a two-step mechanism of binding in which RNA helicases first resolve RG4s and then recruit hnRNP H/F driving their binding to the linear G-rich regions. Thus, our findings refine the model of RBP recruitment by RNA helicases recently proposed[49] by defining the RG4 folding status in the regulatory mechanism. A key question regarding the mechanistic of translational regulation was whether hnRNP H/F simply bind unfolded RG4s or had a function once bound to the linear G-rich regions. The last hypothesis is

**Fig. 5 hnRNP H/F drive genomic instability and therapy resistance. a** Immunofluorescence experiments in LN18 cells using the γ-H2AX, 53BP1 antibodies and DAPI staining. Mean intensities of γ-H2AX and 53BP1 in 2322 cells were plotted; the bottom and top of the box present the first and third quartile, respectively; the band inside the box shows the mean and the whiskers show the upper and lower extremes. Statistical significance was performed on the full cell populations. $n = 2322$ cells examined. Shown is a single representative field from one experiment over $n = 2$ independent experiments. For γ-H2AX: ***$P$-value $= 4.26e-10$, for 53BP1: ***$P < 2.2e-16$ (two-sided Mann & Whitney test). **b** Western blot analysis of γ-H2AX in LN18 cells treated with dose scale of carboxypyridostatin (cPDS) for 24 h. Shown is a representative result from $n = 3$ independent experiments. **c** Quantification of the γ-H2AX levels in LN18 treated with cPDS normalized to GAPDH levels and plotted relatively to the untreated condition. Data are presented as mean values ± SEM of $n = 3$ independent experiments, $P$-value $= 0.0157$ and $P$-value $= 0.0457$ for the 2 μM and 10 μM cPDS treatment respectively (two-sided paired $t$-test). **d** Quantification of DNA repair kinetics by western blot analysis of γ-H2AX after 4 Gy γ-irradiation in LN18 cells treated with control (siCtr) or hnRNP H/F (siH/F) siRNAs. Shown is a representative result from $n = 2$ independent experiments. **e** Plating efficiency assays measuring the cell survival fraction in LN18 treated with siCtr or siF siRNAs and submitted to a radiation dose scale. Data are presented as mean values ± SEM of 6 wells, $P$-value $= 0.0003$ and $P$-value $= 0.0006$ for the 2 Gy and 4 Gy dose, respectively (two-sided paired $t$-test). **f** Quantification of DNA repair kinetics by western blot analysis of γ-H2AX after temozolomide (TMZ) treatment in LN18 cells transfected with an empty plasmid (pICE) or a plasmid expressing Flag-hnRNP H/F. Shown is a representative result from $n = 2$ independent experiments. For all panels, source data are provided as a Source Data file.

supported by our results showing that unfolded RG4s (7dG) still require the presence of hnRNP H/F for their function in translational regulation (Fig. 3g and Supplementary Fig. 8e). While our results suggest that hnRNP H and hnRNP F behave similarly in their interactions (RNA-protein (Fig. 1) or protein-protein (Fig. 4)) and function (Fig. 3) (as previously reported[29,30]), recent data showing that the two factors do not fully share the same set of protein interactors[50], raise important questions about the possibility of differential translational effects discernable at the level of individual mRNAs or in specific translational compartments (cytosol versus microsomes). Finally, DHX36 and DHX9 were shown to stimulate mRNA translation by unfolding RG4s at upstream open reading frames (uORFs)[24]. These results together with our findings support interesting future investigations to determine whether hnRNP H/F are involved in this regulatory mechanism.

In addition to highlighting the possibility that this mechanism may be important for splicing[32] or polyadenylation[29,46], our study extends the functions of hnRNP H/F to translational regulation and assigns to this mechanism a key role in the regulation of genes involved in resistance to treatments in GBM (Fig. 6). Although further work is needed to understand and characterize the full hnRNP H/F translatome, we found that RG4s are overrepresented in hnRNP H/F-binding sites at translational regulatory regions of mRNAs involved in pathways associated to genome instability and DNA damage and that hnRNP H/F bind an important fraction of predicted (Fig. 3b) or experimentally validated RG4s (based on ref. [11]) (Supplementary Fig. 7c). Therefore, we predict that hnRNP H/F drive a substantial part of the RG4-dependent translational regulation and impact on the maintenance of genome integrity. In line with this view, RG4 stabilization by hnRNP H/F silencing or treatment with cPDS, induced the expression of markers of genome instability (Fig. 5). Although it could not be excluded that these effects are associated with the nuclear functions of hnRNP H/F[29,46], we provided evidence that the link between hnRNP H/F and genome stability depends in part on mRNA translational regulation (Fig. 5). Moreover, hnRNP H/F inhibition not only induced but also enhanced chemo- and radio-therapy-induced DNA damage correlated with reduced cell survival (Fig. 5, Supplementary Fig. 10), indicating that targeting the RG4-dependent and hnRNP H/F-sensitive regulatory mechanism sensitizes cancer cells to treatments currently used to treat GBM patients (Fig. 7). Mining GBM TCGA and REMBRANDT data sets (Supplementary Fig. 4) as well as analyzing the protein expression in human glioma protein samples (Fig. 6), we found that hnRNP H/F is increased in GBM and correlates with poor survival, extending the notion of a key role of hnRNP H/F family members in cancer development and progression[63]. Our results support a model (Fig. 7) in

which hnRNP H/F overexpression in GBM coordinately regulate the translation of RG4-containing mRNAs encoding proteins involved in maintaining genome stability and in the response to genotoxic damage. The observation that 74 mRNAs coding for stress response factors are targeted by both hnRNP H/F and DHX36 (Supplementary Fig. 12) opens up new avenues for future research to investigate whether and how these regulations induce adaptive changes crucial for tumor cell survival during treatment and the development of resistance. Our results not only extend the notion of a link between G4 and genomic instability[64] to mRNA translational regulation but also associates it with a role in resistance to treatments in GBM. Given that 1) our results were similar regardless of the GBM cell line (e.g., Fig. 2a, c, d and Supplementary Fig. 5b; Fig. 3g and Supplementary Fig. 8e, Fig. 5d and Supplementary Fig. 10b) or the cancer cell-type (Supplementary Fig. 13), 2) hnRNP H/F were found to be deregulated in many tumors (Supplementary Fig. 14), and 3) hnRNP H/F RG4-containing mRNA targets significantly enriched genes associated to GBM (adjusted $P$-value $= 0.03284$ and $0.001729$ for H and F targets, respectively) but also to other cancers, including breast (adjusted $P$-value $0.033$ and $1.2E-06$) and ovarian cancers (adjusted $P$-value $0.013$ and $1.8E-05$), we propose that the link between hnRNP H/F and cancer mediated by RG4-dependent translational regulation could apply to other cancer cells and tumors, thus making hnRNP H/F a potential target for therapeutic intervention.

Overall, our results support the notion that hnRNP H/F are an essential regulatory hub in GBM networks that drives translational control of RG4-containing genes contributing to GBM progression and response to treatments. Moreover, our RP-MS screen raises interesting future investigations to determine how modulation of RG4 structural integrity impacts cellular functions related to cancer hallmarks.

## Methods

**Cell culture and treatment.** Glioblastoma cells (LN18, ATCC CRL-2610; U251-MG ECACC #;09063001 U87, SIGMA, #89081402-1VL) were grown in DMEM media (4.5 g/l glucose) supplemented with 10% FBS, 2 mM L-glutamine, 100 U/ml penicillin, and 100 μg/ml streptomycin. Cells were tested for mycoplasma contamination by PCR. Cells were incubated/exposed at 37 °C with: 20 μM PDS (Selleckchem S7444) or 20 μM cPDS (Sigma-Aldrich SML1176) or 10 μM PhenDC3 (Polysciences, #26000-1) for the indicated time, 100 μg/ml Puromycin (Sigma P8833) for 1 h, 500 μM or dose scale of TMZ for 24 h, 4 Gy or dose scale of γ-irradiation (Gammacell 40 Exactor).

**Cell transfection.** siRNAs were transfected using the Lipofectamine RNAiMAX (Life Technologies) according to the manufacturer's instructions. In brief, cells were reverse-transfected with 2.5 nM siRNA for 48 h. siRNA oligonucleotides Control (5′-GGUCCGGCUCCCCCAAAUG dTdT-3′), against hnRNP H (5′-GG UAUUCGUUUCAUCUACA dTdT-3′), hnRNP F (5′-GGUGUCCAUUUCAU CUACA dTdT-3′) and DHX36 (5′-GGUGUUCGGAAAAUAGUAA dTdT-3′)

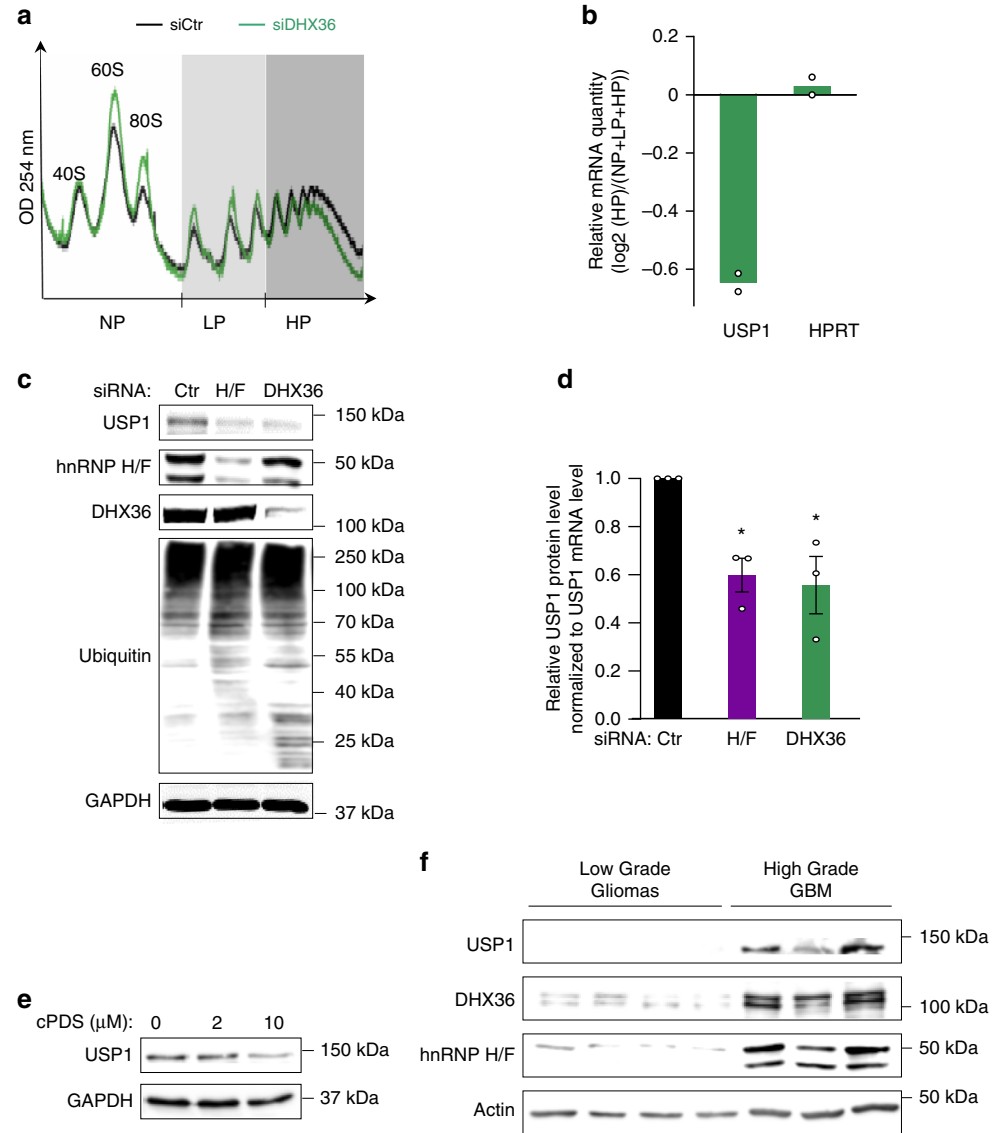

**Fig. 6 hnRNP H/F and DHX36 regulate USP1 translation in glioblastoma cells and tumors.** **a** Polysome profile of U87 cells treated with control (siCtr) and DHX36 (siDHX36) siRNAs. **b** As in **a**, followed by RT–qPCR analysis from pooled non-polysomal (NP), light (LP) and heavy (HP) polysomal fractions, using specific primers for USP1 and HPRT mRNAs, and quantification by analyzing the ratio HP/total mRNAs from n = 2 independent experiments. Source data are provided as a Source Data file. **c** Western blot analysis of USP1 and ubiquitination in U87 cells treated with siCtr, siRNAs against hnRNP H/F (siH/F) or DHX36 (siDHX36). Source data are provided as a Source Data file. **d** USP1 protein levels in **c** were normalized first to GAPDH protein levels and then to USP1 mRNA levels and plotted relatively to the siCtr condition. Data are presented as mean values ± SEM of n = 3 independent experiments, P-value = 0.0291 and P-value = 0.05 for siH/F and siDHX36 respectively (two-sided paired t-test). **e** Western blot analysis of USP1 in LN18 cells treated with carboxypyridostatin (cPDS) dose scale for 24 h. Shown is a representative result from n = 3 independent experiments. Source data are provided as a Source Data file. **f** Western blot analysis of USP1, DHX36 and hnRNP H/F levels in protein extracts from Diffuse Low Grade Gliomas (Grade II) and High Grade GBM (grade IV). Shown is a representative result from n = 3 independent western blot. Source data are provided as a Source Data file.

were synthesized by SIGMA. For DNA plasmid transfections, 3.7 µg of plasmids was transfected in 60 mm diameter dishes using jet-PEI reagent (Polyplus) according to the manufacturer's instructions. For Luciferase mRNA transfections, 250 ng of reporter mRNA was transfected in 48-well plates using lipofectamine 2000 reagent according to the manufacturer's instructions. Cells were subsequently incubated at 37 °C for 48 h or 16 h following DNA plasmid or mRNA reporter transfections respectively, before harvesting and analysis.

**Cell fractionation**. For cell fractionation, cells were gently resuspended in 500 µl of hypotonic lysis buffer (10 mM Tris pH 8.0, 1.5 mM MgCl₂, 10 mM NaCl, 1 mM DTT) and vortexed for 4 s. After centrifugation at 1000 g (4 °C) for 5 min, supernatant (cytosolic fraction) was recovered. Pellet fraction (washed twice with hypotonic lysis buffer) was resuspended in 500 µl lysis buffer A (10 mM Tris pH 8.0, 140 mM NaCl, 1.5 mM MgCl₂, 0.5% NP40, 1 mM DTT). The supernatant (microsomal fraction) was recovered. Pellet-nuclear fraction (washed twice and

resuspended in 500 µl of lysis buffer A) was transferred to a 5-ml round-bottom tube and 50 µl of detergent mix (3.3% (w/v) sodium deoxycholate, 6.6% (v/v) Tween 40) were added. After incubation on ice for 5 min, the supernatant-postnuclear fraction was recovered (perinuclear fraction). The pellet-nuclear fraction (washed with buffer A) was resuspended in 500 µl of lysis buffer A supplemented with 0.1% SDS and sonicated. After centrifugation at 1000 g (4 °C) for 5 min, supernatant (nuclear fraction) was transferred into a fresh tube.

**Mass spectrometry**. Proteins were lysed and denatured in Tris 50 mM pH 8.5 and SDS 2% while disulfide bridges were reduced using TCEP 10 mM and subsequent free thiols groups were protected using chloroacetamide 50 mM for 5 min at 95 °C. Proteins were trypsin-digested overnight using the suspension trapping (S-TRAP) method to collect peptides as described in[65]. Eluted peptides were vaccum-dried while centrifuged in a Speed Vac (Eppendorf). C18 liquid nanochromatography and Mass Spectrometry (MS) nanoflowHPLC and MS analyses were performed at

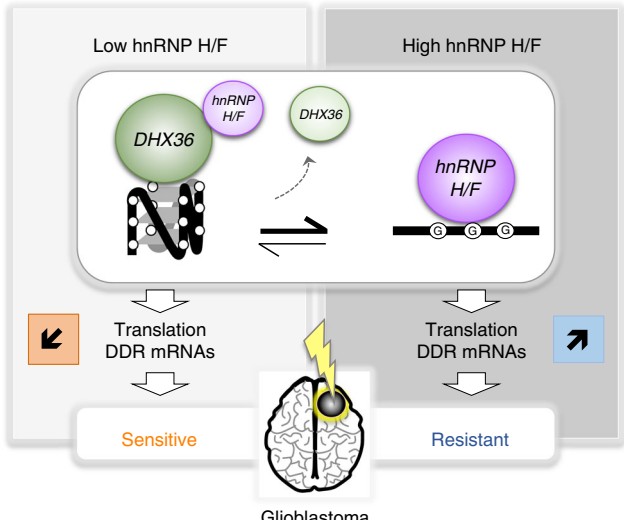

**Fig. 7 Model for the role of hnRNP H/F-RG4 interactions in regulating mRNA translation of mRNAs linked to GBM response to treatments.** hnRNP H/F expression levels in GBM modulate the RG4-dependent mRNA translation impacting the DDR response involved in the response to standard GBM treatments (radiotherapy and chemotherapy). The underlying mechanism involves the binding of the helicase DHX36 that unwinds the RG4s, enabling hnRNP H/F to associate with unfolded RG4s and maintain them linear.

the 3P5 proteomics facility (University de Paris) using an U3000 RSLC system hyphenated to an Orbitrap fusion MS (all from Thermo Fisher Scientific). All mobile phases are made with milliQ-grade H2O produced with a milliQ integral-3 (from Merck-Millipore). Peptides were solubilized in 10 μl of 0.1% trifluoroacetic acid (TFA) and 10% acetonitrile (ACN). 1 μl was loaded, concentrated, and washed for 3 min on a C18 reverse-phase precolumn (3-μm particle size, 100 Å pore size, 75-μm inner diameter, 2-cm length; Thermo Fischer Scientific) with loading solvent containing 0.1% TFA and 2% ACN. Peptides were separated on a C18 reverse phase resin (2-μm particle size, 100 Å pore size, 75-μm inner diameter, 25-cm length; Thermo Fisher Scientific) with a 35-min binary gradient starting from 99% of solvent A containing 0.1% formic acid and ending in 40% of solvent B containing 80% ACN, 0.085% formic acid. The mass spectrometer acquired data throughout the elution process and operated in a data-dependent scheme with full MS scans acquired with the Orbitrap, followed by as many MS/MS ion trap HCD spectra 5 s can fit (data-dependent acquisition with top speed mode: 5-s cycle) using the following settings for full MS: automatic gain control (AGC) target value: 2.10e5, maximum ion injection time (MIIT): 60 ms, resolution: 6.10e4, m/z range 350–1500. For HCD MS/MS: Quadrupole filtering, Normalised Collision Energy: 30. Ion trap rapid detection: isolation width: 1.6 Th, minimum signal threshold: 5000, AGC: 2.10e4, MIIT: 100 ms, resolution: 3.10e4. Peptides with undefined charge state or charge state of 1 or over 7 were excluded from fragmentation, a dynamic exclusion time was set at 30 s. Identifications (protein hits) and quantifications were performed by comparison of experimental peak lists with a database of theoretical sequences using MaxQuant version 1.6.2.10[66]. The databases used were the human sequences from the curated Uniprot database (release June 2018) and a list of in-house frequent contaminant sequences. The cleavage specificity was trypsin's with maximum 2 missed cleavages. Carbamidomethylation of cysteines was set as constant modification, whereas acetylation of the protein N terminus and oxidation of methionines were set as variable modifications. The false discovery rate was kept below 5% on both peptides and proteins. The "match between runs" (MBR) option was allowed with a match time window of 1 min and an alignment time window of 30 min. For statistical analysis, data were imported into the Perseus software version 1.6.1.1[67]. Reverse and contaminant proteins were excluded from analysis. LFQ intensity data were transformed into log2. Samples with at least 3 valid LFQ values of intensity per condition are selected. Imputation was performed on the missing values. Where initial data were insufficient in one condition but enough data in the other condition, the imputation step allowed ratio calculation for all eligible hits (i.e. at least 3 valid values in at least one group). We imputed missing data using a random value comprised in the lowest range of LFQ intensities obtained in MaxQuant with the following settings: 0.3 as gaussian width relative to the standard deviation of measured values, and 1.8 as downshift factor (default perseus values).The proteins were selected as differential if their q-values remained under 0.05 after a permuted FDR test (column x of the Supplementary Data 1). The reproducibility between each replicate was evaluated by hierarchical clustering analysis of protein expression (Euclidean distance) or

Principal Component Analysis (Supplementary Data 1). Log2 of the expression values were used for this analysis.

**RNA chromatography.** 200 μg (WB analysis) or 400 μg (RP-MS analysis) of the U251 cytoplasmic (cytosolic + microsomal fractions (as described in "Cell fractionation" of the Methods section) were precleared with 20 μl of streptavidin acrylamide beads (Thermo Fisher Scientific) in the binding buffer containing 20 mM Tris pH 8, 1 mM DTT, 100 mM KCl, 0.2 mM EDTA for 1 h at 4 °C. For RG4 formation, 1 μg (WB analysis) or 3 μg (MS analysis) of in vitro-transcribed biotinylated RNAs were heated to 95 °C for 5 min in one volume of 1× phosphate-buffered saline supplemented with 2 M KCl and cooled down at room temperature. Biotinylated RNAs were then fixed on 10 μl of streptavidin acrylamide beads by incubation in the binding buffer for 1 h at 4 °C. For PDS/cPDS experiments, 10 μM of PDS/cPDS were then added to the RNA-beads mix and incubated for 30 min. The RNA fixed on beads was then combined to the precleared extracts for 1 or 3 h at 4 °C, for PDS/cPDS and untreated experiments respectively. The beads were collected by centrifugation, washed five times with 1 ml of the binding buffer, resuspended in 30 μl of elution buffer (50 mM Tris pH 8.0, 1% SDS), and boiled for 10 min. After centrifugation, the supernatant was collected. 1 μL was kept for RNA detection and the rest was loaded onto an SDS–PAGE gel and analyzed by western blot or used for MS analysis.

**In vitro transcription.** RNAs used in RNA chromatography experiments were transcribed using the MEGAscript Kit (Invitrogen AM1333) as per manufacturer's instructions. 7.5 mM ATP/CTP, 6.75 mM UTP, 0.75 mM biotinylated UTP (Biotin-16-UTP, Lucigen BU6105H) and either 7.5 mM GTP or 6.75 mM 7-deazaguanine (TriLink N-1044) plus 0.75 mM GTP was used. For luciferase reporter mRNAs, m7G-cap was added using the Vaccinia capping system (M208S NEB) kit according the manufacturer's instructions. To generate the DNA templates to synthetize the luciferase reporter mRNAs, oligonucleotides G3A2 WT, G3A2 Mut, NRAS WT, NRAS Mut were annealed and cloned in the pSC-B-amp/kan plasmid from the Stratoclone Blunt PCR cloning kit, then digested by NheI and purified. All oligonucleotide sequences are available in the Supplementary Table 1. RNA concentration was determined using the Clariostar BMG and software v.5.21 R4, Labtech and MARS Clariostar Analysis Software v.3.20 R2.

**In vitro and in cellulo analysis of translational activity.** For the in vitro translational activity analysis, 100 ng of in vitro transcribed luciferase Renilla reporter mRNAs (WT and 7dG) were preincubated 30 min at room temperature with increasing amount of cPDS. RRL (Flexi Rabbit Reticulocyte Lysate kit) were added to a final volume of 10 μl and the lysates were incubated 90 min at 30 °C. 5 μl of the reaction were used for the luciferase assay. For the in cellulo IRES activity analysis, the U87 or U251 cells transfected with luciferase Renilla and Firefly reporter mRNAs were harvested in 100 μl of Passive Lysis Buffer (Promega). 10 μl of this extract were analyzed with the luciferase assay.

**CD spectroscopy.** For the spectroscopy measurements, RNAs were prepared in buffers containing 10 mM Tris-HCl (pH 7.4), 0.1 mM EDTA in the presence of 100 mM KCl and annealed by heating to 95 °C and then cooling slowly to room temperature. CD of RNAs was determined at 20 °C by a Jasco J-815 spectropolarimeter equipped with a temperature controller. CD spectra ranging from 190 to 350 nm was recorded in a 1-mm path length cuvette, in triplicates, averaged and buffer subtracted.

**Silver staining.** Proteins co-purified by RNA chromatography or present in whole-cell lysates (inputs) were separated by SDS-PAGE and subjected to silver staining using Pierce Silver Stain Kit (Thermo Scientific, 24612) according to manufacturer's instructions.

**Reverse affinity chromatography.** For RG4 formation, in vitro-transcribed biotinylated RNAs were heated to 95 °C for 5 min in one volume of 1 x phosphate-buffered saline supplemented with 2 M KCl and cooled down at room temperature in presence or absence of cPDS 10 μM. hnRNP H/F or DHX36 were immunoprecipitated overnight as described in "immunoprecipitation" of the Methods section. Beads were then washed three times in wash buffer (20 mM Tris HCl pH 8, 100 mM KCl, 0.5% NP-40, 0.4 mM EDTA, 1 mM DTT) and incubated with the in vitro-transcribed biotinylated RNAs in 500 μL of binding buffer (20 mM Tris pH 8, 0.05% NP-40, 1 mM DTT, 100 mM KCl, 0.2 mM EDTA) for 1 h or 2 h when RNAs were incubated with cPDS. After five washes in binding buffer, beads were resuspended in 60 μl of elution buffer (50 mM Tris pH8, 1% SDS), and boiled for 10 min. Immunoprecipitated proteins were analyzed by western blot and biotinylated RNAs were analyzed by urea PAGE followed by biotin detection (as described in "Biotinylated RNA detection" section).

**Surface plasmon resonance.** All binding studies based on surface plasmon resonance technology were performed on BIACore T200 optical biosensor instrument (GE Healthcare) at 4 °C. Capture of the different biotinylated RNA (WT or 7dG) was performed on a Streptavidine (SA) sensorchip in HBS-EP + buffer

(10 mM HEPES, pH 7.4, 150 mM NaCl, 3 mM EDTA and 0.005% surfactant P20 (GE Healthcare). All immobilisation steps were performed at a flow rate of 5 ml/min with final mRNA concentration of 10 µg/ml. Total amount of immobilized ligand was about 1100-1500 RU. The channel (Fc1) was used as a reference surface for all non-specific binding measurements. For binding analysis, cytoplasmic lysates were injected first at 100 µg/ml over the immobilized surface for 2 min at a flow rate of 30 ml/min. Thereafter, the hnRNP H/F antibody was injected at a concentration of 200 µg/ml for 1 min and with the same flow rate settings. The binding of antibodies to molecules captured from lysates by the sensor chips were normalized using BIAevaluation 3.0 software (Biacore AB).

**Biotinylated RNA detection**. Eluates from chromatography experiments were loaded on a 6% UREA polyacrylamide gel and electrophoresed at 4 °C for 1 h at 100 V in 0.5× TAE buffer, and then transferred to either a Biodyne B nylon membrane (Thermo Scientific, 77010) or Hybond-N + nylon membrane (Amersham Biosciences, RPN203B). After cross-link under UV light (UV Stratalinker 1800), signals were probed using the Chemiluminescent Nucleic Acid Detection Module (Thermo Scientific, 89880) according to the manufacturer's instructions.

**Western Blot antibodies**. For immunoblotting analysis, proteins were resolved on 12 or 7% denaturing polyacrylamide gels and were transferred to nitrocellulose membranes. The blots were blocked for 30 min with TBST-5% milk and then probed overnight with primary antibodies against DHX36 (1:1000, Abcam Ab70269), DHX9 (1:1000, Abcam Ab54593), DDX3X (1:1000, Santa Cruz sc-365768), LARP1 (1:1000, Bethyl A302-087A), hnRNP H/F (1:1000, Abcam Ab10689), KSRP (1:500, Bethyl A302-022A), E2F1 (1:500, Santa Cruz sc-251), eIF4A (1:500, Santa Cruz sc-50354), PERK (1:1000, Cell Signaling Technology 3192), Histone H3 (1:1000, Cell Signaling Technology 4499), EEA1 (1:500, Santa Cruz sc-53939), RPS6 (1:1000, Santa Cruz sc-74459), RPL22 (1:1000, Novus Bio NBP1-06069), GAPDH (1:1000, Santa Cruz sc-32233), $\gamma$H2AX (1:1000, Millipore 05-636), Flag (1:1000, Sigma F3165-2MG), USP1 (1:600, ProteinTech 14346-1-AP), Ubiquitin (1:1000, Cell signaling Technology 3936), Puromycin (1:1000, Millipore, MABE343), PARP (1 :1000, Cell signaling 9542), Caspase-3 (1 :1000, Cell signaling 8G10), Anti-Rabbit IgG (1:5000, Ozyme 7074S), Anti-Mouse IgG (1:5000, Ozyme 7076S). The blots were developed using the ECL system (Amersham Pharmacia Biotech) according to the manufacturer's directions.

**Polysomes**. Around $3.10^7$ cells were treated with 0.1 mg/ml cycloheximide (CHX) for 15 min at 37 °C, washed twice with ice-cold phosphate-buffered saline supplemented with 0.1 mg/ml CHX (PBS/CHX), and scraped on ice in PBS/CHX. After centrifugation for 5 min at 200 g, the cell pellet was gently resuspended in 450 µl of hypotonic lysis buffer (5 mM Tris pH 7.5, 1.5 mM KCl, 1.5 mM MgCl₂, 20 U/ml RNaseOUT (Invitrogen, 10777019), 0.1 mg/ml CHX and 10 µl/ml of Protease Cocktail Inhibitor (Sigma, P8340)). The lysate was vortexed for 5 s, incubated on ice for 5 min and 26 µl of 10 % Triton X-100 and 26 µl of 10% sodium deoxycholate were added. After incubation on ice for 5 min, the lysate was centrifuged at 16,000 g for 7 min at 4 °C and a volume of supernatant corresponding to 20 OD₂₆₀ₙₘ was layered on a 11.3 ml continuous sucrose gradient (5-50% sucrose in 200 mM HEPES pH7.6, 1 M KCl, 50 mM MgCl₂). After 2 h of ultracentrifugation at 222,228 g in a SW41-Ti rotor at 4 °C, fractions were collected with an ISCO density gradient fractionation system (Foxy Jr fraction collector coupled to UA-6UV detector, Lincoln, NE). The settings were as follows: fraction time, 62 s/fraction; chart speed, 60 cm/h; sensitivity of the OD₂₅₄ recorder, 0.5. The absorbance at 254 nm was measured continuously as a function of gradient depth; 16 fractions of approximately 0.8 ml were collected. The fractions recovered from the gradient were either analyzed individually or divided into three groups, fractions containing the most actively translated mRNAs, containing more than four ribosomes and called heavy polysomes (HP), fractions containing actively translated mRNAs containing two to three ribosomes, called light polysomes (LP) and fractions containing untranslated mRNAs (non-polysomes (NP)). Equal amounts of RNA from the NP, LP and HP fractions were extracted by using Trizol LS (Invitrogen), analysed by agarose gel and subjected to RT-qPCR analysis to determine the polysomal mRNA distribution. Protein from individual fractions were extracted by using isopropanol precipitation and analysed by western blot.

**SUnSET**. Cells were treated with 10 µg/ml puromycin (Sigma P8833) for 10 min at 37 °C. Cells were washed twice in ice-cold PBS, scrapped on ice in PBS and collected by centrifugation at 200 g for 5 min. Cell were lysed in 50 mM HEPES pH7.0, 150 mM NaCl, 10% Glycerol, 1% Triton, 10 mM Na₄P₂O₇, 100 mM NaF, 1 mM EDTA et 1.5 mM MgCl₂ and 10 µl/ml Protease Cocktail Inhibitor (Sigma, P8340) buffer and puromycin incorporation was analyzed by western Blot.

**Immunoprecipitation**. Cytoplasmic (cytosolic + microsomal fractions (as described in "cell fractionation" of the Methods section)) cell extracts were digested for 1 h at room temperature with Benzonase (Millipore E1014) and DNase I (Thermo Scientific EN0521) and precleared with protein-sepharose beads for 1 h at 4 °C. hnRNP H/F (10 µg, Abcam Ab10689), DHX36 (5 µg, Abcam Ab70269) or BG4 antibodies (0.5 µg, expressed from the pSANG10-3F-BG4 plasmid (Addgene #55756), kindly provided by S. Balasubramanian and purified based on[45]) were

incubated with 20 µl of slurry beads (washed and equilibrated in cell lysis buffer) for 1 h at 4 °C.

Beads were incubated with 1 mg of cell extracts overnight at 4 °C. Beads were washed three times with cell lysis buffer and co-immunoprecipitated proteins were analyzed by western blot.

Purified RNA from mRNP complexes was resuspended in 10 µl of water and 4 µl was reverse transcribed using the RevertAidH Minus First (Thermo fisher) according to the manufacturer's protocol. Subsequently, a 1/5 dilution of cDNA was analyzed by qPCR with the SybrGreen (KAPA KK4605). The mRNA levels associated with these mRNP complexes were then standardized against HPRT mRNA levels (used as a reference) and compared with RNA levels in the IgG control and input sample.

**RT-qPCR**. Reverse transcription (RT) was performed on 1 µg total RNA (quantified with the Clariostar BMG and software v.5.21 R4, Labtech and MARS Clariostar Analysis Software v.3.20 R2) using the RevertAidH Minus First (Thermo fisher) according to the manufacturer's protocol. 12.5 ng of cDNA was analyzed by qPCR with the SybrGreen (KAPA KK4605) using the StepOne software v2.2.2 (Applied Biosystems). Expression of MECP2, PRR5, VEGF, USP1, BABAM1, CCNA2 was standardized using HPRT as a reference, and relative levels of expression were quantified by calculating $2^{\wedge\Delta\Delta CT}$, where $\Delta\Delta CT$ is the difference in CT (cycle number at which the amount of amplified target reaches a fixed threshold) between target and reference. All primer sequences are available in Supplementary Table 2.

**Immunofluorescence**. For the detection of markers of genetic instability, cells grown on coverslips were fixed with 3% paraformaldehyde in PBS for 20 min at room temperature, washed with PBS twice for 5 min, permeabilized with 0.5% Triton X-100/1% normal goat serum in PBS for 15 min, and washed with 1% normal goat serum/PBS three times for 10 min each. The coverslips were then incubated with primary antibodies in 1% normal goat serum/PBS at room temperature for 1 hr using antibodies against $\gamma$-H2AX (JBW301 Millipore 05-636; 1:500) and 53BP1 (Cell Signaling 2675; 1:200). The coverslips were washed twice for 10 min and incubated with goat anti-mouse IgG secondary antibody coupled to fluorescein isothiocyanate in 1% normal goat serum/PBS at room temperature for 1 h. The samples were then washed three times for 10 min each and mounted. For the detection of G4s, cells were seeded in 96-Multiwell plate coated with poly-D-lysine solution. 48 h post seeding cells were pre-fixed with a solution 50% DMEM and 50% methanol/acetic acid (3:1) at RT for 5 min. After a brief wash with methanol/acetic acid (3:1), cells were fixed with methanol/acetic acid (3:1) at RT for 10 min. Cells were then permeabilized with 0.1% Triton X-100 in PBS at RT for 3 min. For RNase treatment, coverslips were incubated with 100 µg/ml RNase A in PBS for 1 h at 37 °C. Cells were incubated with blocking solution (2% milk in PBS, pH 7.4) for 1 h at RT and then with 1 µg per slide of BG4 in blocking solution (2 h at RT). Cells were then incubated with 1:800 of a rabbit antibody against the Flag epitope (Cell Signaling ref# 2368) in blocking solution for 1 h. Next, cells were incubated at RT with 1:500 Alexa Fluor 488 goat anti-rabbit IgG (Life technologies ref# A11008) in blocking solution for 1 h and with DAPI for 10 min. After each step, cells were washed three times for 10 min with 0.1% Tween-20 in PBS under gentle rocking. Cells were visualized at room temperature by using a confocal microscope (Zeiss, LSM780) or using the high-content Operetta High-Content Imaging System (Harmony Imaging 4.8; PerkinElmer). For the high-content analysis, cytoplasmic foci detection and subsequent analyses were performed with Columbus 2.8.2 software (PerkinElmer).

**Plating efficiency, clonogenicity assay**. LN18 glioblastoma cells were transfected with siRNA (siCtr or siF), after twenty-four hours, cells were harvested and plated in 6-well plates at different concentration (500, 750, 1000 cells/well for siCtr and 1500, 2500, 5000 for siF) in duplicate. Twenty-four hours later cells were irradiated with an ionizing radiation scale (from 0 to 4 Gy) using the Gammacell 40 Exactor irradiator (Nordion, Ottawa, Canada) or with TMZ dose scale (from 100 to 500 µM). Cells were then incubated for approximately 10 days until colonies were visible with the naked eye without any joining between colonies. Then, plates were washed and cells were fixed with 10% formalin for 10 min, the formalin was removed and cells were covered with 10% crystal violet oxalate (RAL Diagnostics, Martillac, France) for 10 min, plates were rinsed with water until no additional color comes off the plate. Colonies were then counted to calculate the plating efficiency. Plating efficiency (%) = (number of colonies formed/number of cells plated) × 100.

**GBM tumour sample**. The used protein extracts derived from both low and high grade gliomas were originally processed and used in the article from from Pr JP Hugnot's lab[68]. Total protein lysates in RIPA Buffer (Sigma) (50 mM Tris–HCl pH 8.0, 150 mM NaCl, 1% NP40, 0.5% sodium deoxycholate, 0.1% sodium dodecyl sulfate, 5 mM sodium fluoride, 0.5 mM sodium vanadate, and 1× protease inhibitor cocktail (Roche)) were extracted from 3 GBM (grade IV), and 4 Diffuse Low Grade Gliomas (Grade II: 2 astrocytomas and oligodendrogliomas 2). Tumors samples were obtained from the Montpellier hospital ("biological resource centre", (Collection NEUROLOGIE, DC-2013-2027/DC-2010-1185 /Authorization AC-2017-3055/Research Protocol P487) with patient consent. All the methods used were

carried out in accordance with relevant guidelines and regulations of French Institut National de la Santé et de la Recherche Médicale (INSERM). All experimental protocols were approved by INSERM. The tissues were obtained from patients, who underwent surgical resection. The tissues were processed, classified and graded as described in[69]. The clinicopathological parameters of the patients and tumors are described in the Supplementary Table 3.

**CLIP data analysis**. Reads were trimmed (minimum quality 25, minimum length 18nt) and adapters removed with Trim Galore (https://github.com/FelixKrueger/TrimGalore) with UMIs extracted with UMI-tools (10.1101/gr.209601.116) when needed. Remaining reads were aligned to the hg19 assembly of the human genome with STAR (10.1093/bioinformatics/bts635). Duplicates were collapsed, using UMIs when available. CLIP sites were eventually called with clipper (10.1038/nsmb.2699), using an FDR threshold of 0.05. Background sites were obtained by generating 10000 random sequence sites with length corresponding to the average site length. Sites were annotated for their gene and genomic region of origin with ctk (10.1093/bioinformatics/btw653). RG4 elements were predicted within CLIP sites (extended by 25nt upstream and downstream) and in whole 5′UTR, CDS, and 3′UTR by means of QGRS Mapper (https://doi.org/10.1093/nar/gkl253) with default parameters. RG4 were selected as those predicted RG4 having a score of at least 19. All enrichments were computed with the Fisher exact test, using random sites as background frequencies. Densities per Mb were obtained by dividing the number of sites/RG4 by the total length of the corresponding genomic region obtained from the genome assembly annotations. Gene Ontology analysis was performed in R with the topGO package (Alexa A, Rahnenfuhrer J (2018). topGO: Enrichment Analysis for Gene Ontology. R package version 2.34.0.) and results plotted with REVIGO (http://revigo.irb.hr/, medium similarity threshold). CLIP data for DHX36 was obtained from ref. [25]. Target 5′UTRs, CDS or 3′UTRs were selected as those with at least one significant cluster. Intersections were then performed with the list of hnRNP H/F common targets in the different mRNA regions.

**Data analysis**. Data analyses were performed with Microsoft Excel, Graphpad Prism8, ImageJ v 1.52, R v 3.6.1, RStudio v 1.0.153 and figures were prepared with Microsoft power point, Inkscape v 0.92.4, Gimp v 2.10.18.

**Reporting summary**. Further information on research design is available in the Nature Research Reporting Summary linked to this article.

## Data availability

The mass spectrometry proteomic data have been deposited to the ProteomeXchange Consortium via the PRIDE [1] partner repository with the dataset identifier PXD015609. CLIP data for DHX36, hnRNP H and hnRNP F were obtained from GEO ID GSE105171, E-MTAB-6221, GSE34993, respectively. The REMBRANDT dataset was available at the Betastasis website (http://www.betastasis.com/glioma/rembrandt/kaplan-meier_survival_curve/). The source data underlying Figs. 1d–f, 2a–c, 3a, d–f, g, 4a–e, 5a, b, d–f, 6b, c, e, f are provided as a Source Data file. The source data underlying Supplementary Figs. 1d, 2b–d, 3a–c, 5a, b, 6a–g, 8b–f, 9a, e, 10a, d, e, 11a, b, 13a, d are provided as a Source Data file. All data is available from the authors upon reasonable request. S.M. is the lead contact for correspondence.

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

## Acknowledgements

We are grateful to S. Queille for assistance with immunofluorescence experiments; to M-J. Pillaire for discussion and materials; to A. Willis, for providing the hnRNP I antibody; S. Pautet for assistance with preliminary experiments. We acknowledge members of the S. Millevoi, S. Pyronnet and E. Moyal laboratories as well as N. Puget, D. Gomez and I. Gallouzi for discussions and advices. We thank C. Broussard (LC-MS supervision), M. Leduc (LIMS management), P. Mayeux (proteomic expertise and experimental design) from the 3P5 proteomic facility of the Université de Paris. The Orbitrap Fusion mass spectrometer was acquired with funds from the FEDER through the "Operational Programme for Competitiveness Factors and employment 2007-2013" and from the "Canceropole Ile de France". We are grateful to M. Augustus for providing extracts of tumor samples; L. Ligat (Technology cluster of CRCT) for helping with surface plasmon resonance experiments and confocal imaging; Y. Martineau, S. Shin and A. Olichon for helping with RIP assays using the BG4 antibody; A. De Magis for helping with immunofluorescence assays.

This work was supported by institutional grants (from INSERM, Université Toulouse III - Paul Sabatier, CNRS) and by funding from LNCC (Ligue Nationale Contre le Cancer), ARC (Association pour la Recherche contre le Cancer), Emergence Cancéropole GSO, Laboratoire d'Excellence TOUCAN (ANR11-LABX) and ANR (ANR-17-CE12-0017-01). MLB was supported by the Midi-Pyrénées Region/INSERM, PH by ANR (ANR-17-CE12-0017-01) and LD by MENRT.

## Author contributions

S.M. conceived the project. S.M. and A.C. designed and supervised the experiments. P.H. and M.L.B. performed most of the experiments, together with A.C., with assistance by L.D. and C.H. A.C. made all the figures. J.G. performed TCGA analysis. J.P.H. provided GBM samples. F.G. designed and supervised the proteomic analysis. A.A. performed proteomic analysis and statistical data treatment. E.D. performed bioinformatic analysis. G.C. performed CD experiments. S.M. wrote the manuscript with input from A.C., M.L.B., P.H., L.D., E.D. and F.G.

## Competing interests

The authors declare no competing interests.
