## [Peer Review File · Nature Communications]

Reviewers' comments:

Reviewer #1 (Remarks to the Author):

In this manuscript, the authors started by finding using proteins binding a rG4 (i.e. G3A2) by mass spectrometry. Among the 370 hits, they keep hnRNP F/H for the further analysis. The localization of hnRNP F/H was shown to be at the active translational sites. To validate that hnRNP F/H had a role in translation, they did some knock-down with siRNA targeting hnRNP F, hnRNP H or both of them. They also predict G4 in their mRNA target to see more the impact of the silencing. They find out that hnRNP F/H seems to bind to DNA damage repair mRNA containing pG4. Subsequently they demonstrated that hnRNP F/H needs the action of DHX36 to bind RNA while the contrary is not true. This light a possible mechanism : DHX36 unfolds the rG4 and then hnRNP F/H comes to let the rG4 unfolded. Finally, they tried to find out the impact of hnRNP F/H-rG4 on the translation regulation on the DDR. Those results suggest that hnRNP H/F had a role on the control of the cell response to DNA damage by regulating the translation of some DDR proteins. Taken together, these results may allow to think that hnRNP F/H can be used as a therapeutic agent to improve the effect of traitement of gliomas cancer.

Overall this manuscript is well written and results are supported by their experiments. However, I am not sure if the impact is suffisant for Nature Communication. hnRNP F/H has been reported previously to bind rG4 and the cooperative mechanism of a RBP and the helicase DDX36 is not new (see ref. 6 of the manuscript, inbteraction with Aven). Moreover, the mechanism involving the DHX36 and HNRNP F/H need further investigation.

Following are some comments to improve the manuscript for further resubmission:

1. The author should add a schema of the proposed mechanism.
2. For some experiments there are no explanation about the use a cell line and not others.
3. The authors do not distinguish hnRNP F and hnRNP H in their results, are there any hypothesis on what could be their different roles ?
4. Is this study applicable to other cancer types ? Like, is their other cancer with hnRNP over-expressed ?
5. Line 166 authors mention a threshold of 1.5-fold enrichment but did not explain why. Is it because it's what is usually used ? Is it arbitrary ?
6. In Material and methods, in cell culture and treatment they did not mention the temperature.
7. Even if results are strong, maybe the authors could add an experiment to show that rG4s are folding (for G3A2 of USP1, CCNA2, etc).
8. It would be interesting to show the RNA pull-down with G3A2 Mut in Figure 1B to really show that there is a difference between 7dG and Mut.
9. The authors should work on the presentation of Figure 1E.
10. At line 251, the authors mention that HNRNP H/F was enriched un RE, referring to figure 2A, but there is no mention of RE in figure 2A, only cytoplasm, nuclear and microsomal.
11. Resolution of Figure 2C must be corrected.
12. Figure 4 Ci and Di should be put in supplementary data since the part ii bring all the

information we need.

13. There is no mention of Figure 4D in the manuscript.

14. Figure 6 Bii is not consistent with previous data, should the authors put this in supplementary data since that is what they did with figure 3D?

Reviewer #2 (Remarks to the Author):

In the manuscript "hnRNP H/F drive RNA G-quadruplex mediated translation linked to genomic instability and therapy resistance in glioblastoma", the authors performed an affinity purification assay coupled to mass spectrometry to identify novel RNA G4 interacting proteins. Their unique approach is to use a control oligonucleotide bait in which the guanines are replaced by 7-deaza-guanine, which prevents Hoogsteen base pairing and consequently G4 formation. In this approach they have identified hnRNP and hnRNP F as proteins not interacting with folded but with unfolded G4s motifs. In subsequent experiments they aimed to characterize in vivo the biological function of these proteins at G4 sites. As a final approach they use the gained knowledge to shed light on GBM treatment and therapy. Despite the presented experiments it is not clear if hnRNP H/F simply binds G-rich regions or has a function in keeping G4 motifs unfolded. Due to many overinterpretation of the data, multiple comments arise which needs to be addressed.

See below my specific comments

1. The idea to use 7-deaza-guanine is very interesting in an affinity purification strategy. But many open questions arise from this experiment: They identified 370 G4 interacting proteins, of which some were novel. Detailed information on these proteins are needed in the main text. How many where identified in all four biological replicates (this information is in the Suppl Table 1, but needs to be stated in the text), how many of those have been previously identified in the literature? Only 27, which seems rather a low number of joined proteins? How about the proteins identified in addition to Heddy et al? Does this mean the other proteins identified in the literature are not correct? Are those identified in the current approach? How can the authors be sure that their protein list is more correct than the published version? Interestingly, the authors claim that most (238) of their proteins are not known RBPs. Does this mean they were not identified in the two large mRBPs screen (Hentze lab: Castello et al 2012 Cell or Landthaler lab: Baltz et al 2012 Mol Cell) or are listed in a census of human RBPs (Tuschl lab: Gerstberger et al 2014, Nature Reviews)? Would that mean most G4 interacting proteins are not mRNA binding proteins? Also, it is experimentally not clear, why many nuclear proteins where identified albeit they used a cytoplasmatic pull down approach. Is this the reason why no significant overlap to known RBPs was identified? To confirm RP-MS results they did Western blot analysis of a few known G4 interacting proteins by the same approach. To confirm robustness of assay, it is more appropriate to perform an inverse IP, meaning pull-down of the protein and detection of the RNA.

2. Figure 1: Figure 1A, should be in Suppl, B, is very poor quality, what is in loaded in 2 and 3? Why is the marker not included in the gel or is this lane 2? D the rational around these figure is not clear. E how where the ratios calculated: e.g. first lane DHX36 should be 1.5 enriched in WT compared to 7dG, I see not band in 7dG, however in DHX9 nearly the same ratio is blottet but the band intensity looks identical in G4 and controls? LARP1, which was identified previously as a G4 binder here shows in Figure 1E robust binding to 7dG as well as the mutG4? KSRP should be a control for a protein binding to unfolded G4s, but also here the calculations of the ratio is strange, nearly the same ratios are written as for the hnRNP, but band intensities vary. Most importantly they draw the conclusion that because the hnRNP band is stronger for the 7dG that it binds better to 7dG than to G4. Subsequent in vitro experiments are needed to support this conclusion. Meaning CD spectra of G3A2 constructs as well as binding analysis (e.g. MST, EMSA) are essential to state such a strong claim F the experimental set up is not clearly stated. in the remaining manuscript the authors use cPDS, why did they perform these experiments with PDS? the choice

of hnRNP I as a control is not clear, it is mainly nuclear and why does it bind at all because the pyrimidine do not alter in the constructs and there are no poly pyrimidines in the constructs? Mainly a reverse IP is essential to verify this interaction. Pull down of the proteins and detection of the G3A2.

3. Among many proteins they focused on hnRNP H/F, which might be linked to GBM. Although many indirect evidences were pointed out by the authors no direct evidence was shown. Figure 2A Nevertheless, their approach to check for the localization in different GBM is valid and makes sense, however why are the authors not surprised to find it mainly in the nucleus? "But co-distribute with proteins associated with active translation (lane 50)" Do they mean cytosol fraction? Why is there no fractionation control for the cytosol fraction like tubulin? B where are the fractions 1-7? This would give a real idea how much of the total cellular hnRNP H/F is interacting with the ribosome and how much is unbound or in low-molecular complexes. If the puromycin just makes the 80S more abundant, is the shift simply based due to changes in polysome profiling? C why is PDS and not cPDS used? 60S and 80S fractions are stronger, shift towards these regions of proteins might be due to G4 but also could be simply be due to indirect effects. PDS alters transcription and translation at many levels and observed effects are over- interpreted. Lines of the UV profile are blurred compared to other shown polysome profiles

4. Figure 3: the aim here was to re-analyze published datasets and set them into context of new hypothesis. Directly: are hnRNP H/F binding sites G4 regions. For this they used predicted G4 by QGRS mapper, why did the author not use experimental identified G4 regions (Kwok et al., 2016)? lane 285.....bound an important fraction of predicted RG4.. here a more scientific number is essential to validate these analyses. Is this more than expected by chance? What are random control regions? Are the UTRs meant by this? It is not too striking that more is the UTRs, because they are more G-rich than the CDS. Figure 3 C GO term analysis is a good idea but analysis needs to be validated experimentally. I agree with the authors that there is a link to DDR but also to many other pathways, the link to GBM is weak and C belongs to the supplement figures. D the experiments are well designed, but the conclusion raised by the authors is too strong. They interpreted the data (lane 316ff) that hnRNP H/F depletion induced a significant modification of mRNA association with translating polysomes...this is consistent with the notion that RG4s can either activate or suppress mRNA translation. First the observed effects are super minor and can be simply be due to experimental variations? Where is the error bar and statistical tests? However, if there are minor changes, these can be RG4 independent and experimental proof are essential to support their strong conclusion.

5. Figure 4: formatting errors in C. Why are there not statistical validations in all other panels? Western blots of C and D should be moved to supplement

6. Figure 5: why do they switch to cPDS? The experiments are valid, but the conclusion drawn from then is too strong. That G4 stabilization induces DNA damage is known and anticipated, an addition of ligands changes transcription and translation at multiple regions. A reporter assay or global analysis combining targets for hnRNP H/F and DHX36 and transcriptional changes in genes associated with DNA repair are needed.

7. The paper and especially the discussion would benefit if the Wolfe et al paper would be included

Minor

- Lane 148: the term RG4 regions is misleading. Proteins unfold or unwind RG4 but not the regions.
- Major language and formatting changes are required e.g. lane 166 (to to), lane 859
- Problems in labeling of figures and Supps. E.g. lane 188-189, which is not Supp, it should be 1e. Figure 4D is not mentioned in the main text
- KSRP named in text, KhSRP in the table, make uniform
- In methods its odd to change in one sentence between two hundred micrograms and 400 hundred micrograms (lane 753), in the later case the "hundred" is also redundant
- Western blot procedure is missing, only names of antibodies not sufficient

Reviewer #3 (Remarks to the Author):

The manuscript by LeBras et al describes a very complete set of experiments that, for the first time, identified confidently proteins that bind G-rich RNA sequences either in the folded G-quadruplex form or single-stranded form. The presence and role of RNA G-quadruplexes in cells is currently controversial and it has been suggested that G-quadruplexes exist as transient regulatory structures that are in competition with RNA binding proteins. Furthermore, the authors characterize the splicing factors hnRNP H/F and show that they bind single-stranded G-rich RNAs and are involved in translation, in cooperation with the helicase DHX36, to control a subset of genes involved in DNA damage response. Finally the authors demonstrate that overexpression of hnRNP H/F contributes to glioblastoma progression.

Overall, the manuscript is well written, the study is well designed and provides valuable insights into the cytoplasmic role of the hnRNP H/F family of proteins. This work is highly appropriate for Nature communication.

Reviewer #4 (Remarks to the Author):

This study aims to determine proteins that associate with unfolded RG4s compared with folded RG4s. An important family of proteins hnRNP H/H were studied along with various RNA helicases and their function of RG4 mediated translation was explored. A variety of techniques were employed including quantitative proteomics and polysome profiling.

The study is interesting and lends more support to the controversy that has surrounded the presence and importance of in vivo RG4 species.

My concerns about this paper are focussed almost entirely on the quality of the data presented.

1. the proteomics data is unclear. The authors state that they find 328 RG4BPs and 42 G-rich BPs. In the supplementary table all proteins are present in all enrichments, irrespective of the RNA input. Firstly the number of interacting proteins seems high, especially for RG4s, and secondly I would have expected some unique interactors that would manifest themselves as present in only one of the conditions. How was data imputation performed on missing values and was this the reason for the lack of presence/absences?

Also the authors have chosen a cut-off of 1.5 fold –how is this justified? Are enough replicates run to justify this threshold?

Of the new RBPs identified, how many of them have been identified in three recently published papers that use new approaches to expand the RBP data bases (PTEX1, XRNAX 2 and OOPS 3 papers)?

The proteomics data seem incomplete, and I would have liked to have seen data for proteins binding to the mutated sequence, UCUGCGAAGUGAAGUGAAGCGAUC, to give more credence to the rest of the proteomics data and filter further background interactions not specific for the structures under consideration.

2. the authors cherry picked proteins of interest from their proteomics data with no clear reasons why other in the list should be ignored and then carry out pulldowns and westerns. The data presented in figure 1E are not compelling. The statement in the text (line 193) saying that the pulldown of the helicases showed 'less' interaction G3A2 7dG than with G3A2 WT is not satisfactory. Where are the replicates and error bars that support this statement?

3. all the conclusions made about the data portrayed in figure 2 would be much better supported by the use of replicates. In particular, the western blot displayed in figure 2C is of poor quality, and based on it I do not accept the conclusions made.

4. In general all polysomal profiling data is badly represented. In figure 3 (and figure 6) the lack of replicates make the subtle changes in distribution of proteins among the different populations of monosomes and polysomes difficult to accept.

5. It is not clear to me why the immunoprecipitations described figure 4 were not carried out in the C and M fraction separately, as interaction of hnRNP H/F with the translational machinery may be

different in the cytosol compared with the ER.

6. I do not agree with the statement from line 355 onwards about the preferences for hnRNP H/F and DHX36 for folded or unfolded RG4 and the dependency on DHX36 by hnRNP H/F for binding to its targets. The data on which this statement is made is mainly qualitative (i.e. pictures of westerns) that without replication, makes the statement extremely speculative.

1 Urdaneta, E. C. et al. Purification of cross-linked RNA-protein complexes by phenol-toluol extraction. *Nature Communications* 10, 990, doi: 10.1038/s41467-019-08942-3 (2019).

2 Trendel, J. et al. The Human RNA-Binding Proteome and Its Dynamics during Translational Arrest. *Cell* 176, 391-403.e319, doi: <https://doi.org/10.1016/j.cell.2018.11.004> (2019).

3 Queiroz, R. M. L. et al. Comprehensive identification of RNA-protein interactions in any organism using orthogonal organic phase separation (OOPS). *Nature Biotechnology*, doi: 10.1038/s41587-018-0001-2 (2019).

NCOMMS-19-11272A : *hnRNP H/F drive RNA G-quadruplex-mediated translation linked to genomic instability and therapy resistance in glioblastoma*

By Herviou, LeBras et al.

Detailed Response to Reviewers' Comments:

Reviewer 1

We are grateful to the Reviewer for his constructive comments and for having pointed out issues that we addressed in the revised version.

1.0 hnRNP F/H has been reported previously to bind rG4 and the cooperative mechanism of a RBP and the helicase DDX36 is not new

We are grateful to the Reviewer for having pointed out the aspect of novelty and originality of this manuscript in relation to previous work, so that we can further highlight them in the revised version.

Concerning the novelty, while it is known that hnRNP H/F binds RG4s (and our previous work (1) was groundbreaking in showing this result), its translational role mediated by RG4s, the underlying regulatory mechanism, the widespread of this regulation and the physiopathological consequences remain largely unknown to date. Moreover, our work goes beyond the simple demonstration that hnRNP H/F binds RG4s; not just because it reports for the first time a proteome-wide study on RBPs binding folded/unfolded G-rich sequences, but also because it addresses the fundamental question of the machinery that would be responsible for maintaining RG4s in a linear conformation *in cellulo*, as recently proposed by Guo (2) and more recently refined by Yang (3).

We believe that this notion was stressed in both the introduction and the beginning of the discussion (submitted version).

With regards to the RBP-helicase cooperation, Aven was shown to cooperate with DDX36 to enhance the translation of two mRNAs linked to leukemia (4), but several elements indicate that our work extends beyond this study. Indeed, unlike Aven, hnRNP H/F 1) is found in all the proteomic-wide studies identifying RG4-binding proteins (e.g. (5-8), 2) is a canonical RBP found in all the RBP catalogues (new **Suppl. Table S2**) (9, 10), 3) is abundantly and ubiquitously expressed in most cancer cells and tissues (**Suppl. Fig. S12-13**). Therefore, the findings described here show a general mechanism of RG4-dependent regulation susceptible to have a broader impact. Concerning the widespread of the RBP-RG4 interaction in translational regulation, this notion (supported by our work in Fig. 3) is missing in Thandapani's study (4). In our manuscript, this information is endorsed by both the analysis of the RG4 present in hnRNP H/F binding sites and by the fraction of hnRNP H/F-bound

RG4s over to all RG4 predicted (or experimentally validated) (**Fig. 3 and Suppl. Fig. S7**). Furthermore, with regards to the cooperation between the RBP and DHX36, our work refines the model of recruitment proposed by (4), by defining the folding status of the RG4 in the regulatory mechanism and, above all, links this mechanism to genomic instability, a characteristic of most cancer cells.

We modified the text in the results and discussion sections to further point out the relevance of this study.

1.1 The author should add a schema of the proposed mechanism

We have now added a schema recapitulating the main results of this work in Fig. 7.

1.2 For some experiments there are no explanation about the use a cell line and not others.

Throughout the manuscript, several GBM cell lines with distinct characteristics in terms of mutations and sensitivity to chemo-radio treatments were used to define whether the effect was cell line specific. Based on the results obtained, which did not change according to the lines used (see for example **Fig. 2 A, Fig. 2C and Suppl. Fig. S5B, Fig. 3F and Suppl. Fig. S8D, Fig. 5C and Suppl. Fig. S10B**), we came to the conclusion that the observed effects can be generalized to GBM cells regardless of the status of the mutations or the sensitivity to treatments. This conclusion has been added to the discussion section of the manuscript.

1.3 The authors do not distinguish hnRNP F and hnRNP H in their results, are there any hypothesis on what could be their different roles?

Indeed, we did not observe a difference between hnRNP H and hnRNP F, either in protein-RNA (**Fig. 1 and Suppl. Fig. S1-3**) and protein-protein interactions (**Fig. 4 and Suppl. Fig. S9A**) as well as at the global (**Suppl. Fig. S6**) or the mRNA specific (**Suppl. Fig. S11**) functional level. This result, which is in agreement with previous studies (1, 11, 12), suggests some redundancy between the two factors. Our data (**Suppl. Fig. S6**) showing that the effect of the combined depletion of the two factors is greater than that obtained by the individual loss of each factor supports this redundancy. Recent data showing that the partners of hnRNP H and hnRNP F are not strictly overlapping (13) suggest that the regulatory mechanism may involve complex and probably different series of RNA-protein and protein-protein interactions that could result in differential effects discernable at the level of individual mRNAs. We included this response in the discussion section.

1.4 Is this study applicable to other cancer types ? Like, is their other cancer with hnRNP over-expressed ?

To address these questions, we explored transcriptomic data (RNA-seq) across all cancer cohorts using the firebrowse interface. This analysis highlighted that both proteins were deregulated in several cancers, suggesting that our main findings could be extended to other cancer cells and tumors (**Suppl. Fig. S12**). These results are in agreement with a previous study analyzing the expression of 800 RBPs which showed that hnRNP H is one of the most altered proteins over 9 cancers (14). To further explore this question, we determined whether the role of hnRNP H/F in RG4-mediated translation is cell-type specific. The results showed that the effects observed in GBM cell lines were replicable in colon cancer cells (**Suppl. Fig. S13**). Following the suggestions of Reviewer 2 (see § 2.4.b), we also determined whether mRNAs containing RG4 and interacting with hnRNP H/F enriched genes associated with diseases in the OMIM database. This analysis revealed a significant association with GBM but also with other cancer diseases such as breast and ovarian cancer. The discussion has been modified to take into accounts these results.

1.5 Line 166 authors mention a threshold of 1.5-fold enrichment but did not explain why. Is it because it's what is usually used ? Is it arbitrary ?

We chose an arbitrary threshold value. We added this information in the text

1.6 In Material and methods, in cell culture and treatment they did not mention the temperature.

We have rectified this omission in the new version of the manuscript.

1.7 Even if results are strong, maybe the authors could add an experiment to show that rG4s are folding (for G3A2 of USP1, CCNA2, etc).

This issue being indeed crucial, we addressed it using three different techniques. First, to provide evidence of the G3AA RNAs (WT and 7dG) G4 folding status, we performed circular dichroism (CD). We obtained CD spectra similar to that previously reported for the G3A2 RG4 (2), with peaks characteristics of parallel RG4s (**Suppl. Fig. S1C**). A modification of the intensity of these peaks was observed when comparing the G3A2 WT and 7dG in the presence of K⁺. These results indicated that the G3A2 WT RNA, but not the 7dG version, did form RG4s. The low concentration of available RNA (limited by the synthesis of 7dG RNA) explains why the CD spectra are blurred. To further address this question *in cellulo*, we performed RIP experiments using a specific antibody that recognizes RG4s in living cells (BG4 (15)). The results presented in **Fig. 3C** showed that all the mRNAs considered in this study, including USP1, have the ability to form RG4 structures. Overall, these data support the conclusion that the G-rich sequences bound by hnRNP H/F form RG4s *in vitro/cellulo*. The text has been revised accordingly.

1.8 It would be interesting to show the RNA pull-down with G3A2 Mut in Figure 1B to really show that there is a difference between 7dG and Mut.

We carried out the experiments suggested by the Reviewer (**Supp. Fig. S1**), which showed distinct overall patterns of pulled-down proteins for the three RNAs. The text has been revised accordingly.

1.9 The authors should work on the presentation of Figure 1E.

We agree with the Reviewer that the WT/deaza ratio added next to the western-blot and corresponding to MS data was misleading (see also §2.2.C). For clarity's sake, in the new version, we have omitted this ratio and added the quantification of the RNA-affinity chromatography results in **Fig. 1C** (revised manuscript), showing significant modification of RBP recruitment depending on the RNA used.

1.10 At line 251, the authors mention that HNRP H/F was enriched un RE, referring to figure 2A, but there is no mention of RE in figure 2A, only cytoplasm, nuclear and microsomal.

We thank the referee for pointing out the mismatch between the text and the **Fig. 2A**. The text has now been modified accordingly.

1.11 Resolution of Figure 2C must be corrected.

To answer the Reviewers 1 and 2 who pointed out the poor quality of the figure as well as that in some figures we use PDS and in others cPDS (see § 2.2.e), we carried out the co-sedimentation experiments with U251 extracts (**Figure 2C**), in the presence of cPDS. To show that this effect is cell line-independent, we repeated this analysis using GBM U87 cell extracts (**Suppl. Fig. S5B**). The results obtained were similar when comparing both the two cell lines and the treatment with the stabilizing ligand (PDS versus cPDS) (**Fig. 2C** and **Suppl. Fig. S5A,B**).

1.12 Figure 4 Ci and Di should be put in supplementary data since the part ii bring all the information we need.

We moved these figures in supplementary data (**Suppl. Fig. S9C,D**), as requested.

1.13 There is no mention of Figure 4D in the manuscript.

We thank the Referee for this omission; we modified the text to refer to **Fig.4D**.

1.14 Figure 6 Bii is not consistent with previous data, should the authors put this in supplementary data since that is what they did with figure 3D?

We followed the Reviewer's suggestion to match the representation of **Fig. 3D** and **6B** (revised version). In order to address also the Reviewer 2's concerns

(§ 2.4.c), we have kept only the RT–qPCR analysis from pooled NP, LP, HP fractions in the main text and moved the RT–qPCR analysis from individual fractions in Supplementary Data (**Suppl. Fig. S8B, S11A**, the revised version).

+++++

Reviewer 2

We are grateful to the Reviewer for highlighting the novelty brought by our work and for having suggested several experiments that we performed. We believe that altogether the data provided reinforced our conclusions.

2.0 Despite the presented experiments it is not clear if hnRNP H/F simply binds G-rich regions or has a function in keeping G4 motifs unfolded.

The point raised by the Reviewer is of great importance to deepen our understanding of RG4 remodelling by RBPs. To address this issue, we have synthesized capped and polyadenylated RNA reporters in which the translation of the luciferase ORF is driven by the USP1 mRNA UTRs containing a RG4 wild-type or mutated (USP1 RG4 WT and USP1 RG4 Mut, respectively). The WT RNAs were also transcribed in the presence of 7dGs to obtain the USP1 7dG RNAs. Then, these three USP1 RNAs were transfected into GBM cell lines in which hnRNP H/F were depleted or not. We reasoned that if hnRNP H/F simply bound the G-rich sequences to keep them unfolded (hypothesis A), we should have observed that the depletion of hnRNP H/F had no effect on the translation of the USP1 7dG RNA. By contrast, if the unfolding of the RG4 alone was not sufficient and hnRNP H/F were necessary once RG4 is unfolded to perform its function as a translational regulator (hypothesis B), we would have observed that the loss of hnRNP H/F induced a change in the translation of USP1 RNA 7dG. The **Fig. 3F** (revised version), showed that hnRNP H/F silencing in U87 cells modified the translation of the USP1 7dG RNA, validating the B hypothesis and suggesting that hnRNP H/F have a function in translation beyond simply binding to G-rich sequences to keep them unfolded. These results were replicated in U251 GBM cells, showing similar results (**Suppl. Fig. S8D**, revised version). As expected, in both cell lines, hnRNP H/F silencing did not affect the expression of the reporter in which the RG4 was mutated (**Fig. 3F** and **Suppl. Fig. S8D**). Importantly, our results highlighted a significant differential effect of hnRNP H/F silencing in U87 cells between the translation of USP1 WT and 7dG RNA reporters, with the latter being more affected by hnRNP H/F loss. These results, which were validated also in U251 GBM cells, mirrored the ability of hnRNP H/F to bind the two RNAs. This suggests that the dynamic equilibrium between RG4 and linear G-rich *in cellulo* results in low binding of hnRNP H/F to RG4s but, when preventing RG4 from folding, hnRNP H/F strongly binds the G-rich RNA to potentiate translation. This interpretation, which is in agreement with recent data (3), provides an explanation of the significant but subtle effects of hnRNP H/F depletion on the translation of its targets (**Fig. 3D** and **Suppl. Fig. S8B**). These hypotheses have been added in the text.

2.1.a They identified 370 G4 interacting proteins, of which some were novel. Detailed information on these proteins are needed in the main text. How many were identified in all four biological replicates (this information is in the Suppl Table 1, but needs to be stated in the text), how many of those have been previously identified in the literature? Only 27, which seems rather a low number of joined proteins? How about the proteins identified in addition to Heddy et al? Does this mean the other proteins identified in the literature are not correct? Are those identified in the current approach? How can the authors be sure that their protein list is more correct than the published version?

The questions raised above concern **Fig. 1 D** in which we analyzed our RP-MS results in relation to those of Herdy (8). We chose this dataset because it was the only study that identified cytoplasmic proteins binding RG4s (additional studies mentioned in the introduction used total extracts (5-8)). This comparison, like any comparison between large-scale studies, highlights qualitative and quantitative differences that do not allow us to firmly conclude which of the two analyses is more accurate. Even if the approach to identify RG4-binding proteins is similar, the two analyses differ in: **1) The cancer- and cell-type** (in Herdy, cervix epitheloid carcinoma, while this study, glioblastoma astrocytoma,), **2) The RG4-forming sequence** (this study: G3A2 as in Guo's study (2), Herdy: NRAS RG4). RG4 structuration should result in a parallel RG4 in both but the linear sequence is different. **3) The RNAs used for the comparison** is different: in Herdy, the WT was compared to the mutated (3G-less), while in our study, the WT has been compared to an identical sequence unable to form RG4s. **4) MS methodology**: LC-MSMS with two replicates allowing a qualitative analysis in Herdy; HCD-MS/MS with four replicates allowing a quantitative analysis in our study. All these arguments support the conclusion that the comparison has some limitations and is not meant to provide a conclusion about the correctness of the data.

The revised version now includes detailed information on the 343 novel RG4-binding proteins, compared to Herdy (**Suppl. Table S1**). This high number of novel high-confident RG4 proteins can be explained by the fact that, unlike Herdy, we used a quantitative RP-MS approach with 4 replicates, thus increasing the significance level of our protein interactors. As requested, we added a gene pathway analysis of the significantly enriched functional group for the novel proteins indicated in **Fig. 1B**.

Concerning the number of proteins in the 4 replicates, we added in **Suppl. Table S1** the "R" column reporting valid values before imputation. We found 237 proteins in the 4 replicates, i.e. 60% of the identified proteins by MS.

2.1.b Interestingly, the authors claim that most (238) of their proteins are not known RBPs. Does this mean they were not identified in the two large mRBPs screen (Hentze lab: Castello et al 2012 Cell or Landthaler lab: Baltz et al 2012 Mol Cell) or are listed in a census of human RBPs (Tuschl

lab: Gerstberger et al 2014, Nature Reviews)? Would that mean most G4 interacting proteins are not mRNA binding proteins?

We agree with the Reviewers (see also § 4.1.c) that it was inappropriate to choose only one study to assign the identified proteins to the RBP category, especially because several large-scale studies (using different identification methods and different cell lines) gave not quite overlapping RBP catalogues. We therefore re-analyzed our data by taking into account the studies indicated by the Reviewer (16-20). We have now added a Table in which we report the different studies and their intersection with our data (**Suppl. Table S2**). In addition, we have modified the Venn diagram by intersecting our RP-MS data with a list of RBPs present in at least two of these aforementioned studies (Fig. 1B, revised version). This analysis showed that the most of the identified RG4 proteins (260 out of 370) were previously identified as RBPs by transcriptome-wide protein-RNA interaction studies. This ratio is comparable to that reported by Herdy (8).

The text in the new version of the manuscript has been modified, accordingly.

2.1.c Also, it is experimentally not clear, why many nuclear proteins were identified albeit they used a cytoplasmatic pull down approach. Is this the reason why no significant overlap to known RBPs was identified?

We believe that reason is that many RBPs are able to shuttle between the nucleus and the cytoplasm and are likely to play many roles in both compartments (20, 21). This is the case of hnRNP I (22-24) (see also § 2.2f). The explanation does not lie in the overlap with RBPs that has been revised, now showing a significant proportion of RBPs in our RG4-binding proteins list.

2.1.d To confirm RP-MS results they did Western blot analysis of a few known G4 interacting proteins by the same approach. To confirm robustness of assay, it is more appropriate to perform an inverse IP, meaning pull-down of the protein and detection of the RNA.

As suggested by the Reviewer, we pulled-down hnRNP H/F and DHX36 and detected the bound RNAs (WT or 7dG). Our results (in **Suppl. Fig. S3B,C**, revised manuscript) confirmed the effects obtained with affinity chromatography (**Fig. 1C**, revised version), reinforcing the conclusion that the ability of hnRNP H/F and DHX36 to bind RG4s depend on the RNA conformation.

2.2.a Figure 1: Figure 1A, should be in Suppl, B, is very poor quality, what is in loaded in 2 and 3? Why is the marker not included in the gel or is this lane 2?

As suggested, we moved **Fig. 1A** in supplementary data (**Suppl. Fig. S1B**, revised manuscript). To improve this figure and address the concern of Reviewer 1 (raised in § 1.8), we repeated this experience and indicated what has been loaded in each lane (**Suppl. Fig. S1D**, revised manuscript).

2.2.b Fig. 1D the rationale around these figures is not clear

The rationale is to analyze the proteins from RP-MS to define if they bind RNA but also whether they have been already found in a similar study with cytoplasmic extracts (8). In addition, the intersection with the work of Herdy points out that some proteins identified as G4-RBPs in (8) are indeed G-rich RBPs. We chose this representation to intersect different information relevant to the study and used for the discussion.

2.2.c Fig. 1E how were the ratios calculated: e.g. first lane DHX36 should be 1.5 enriched in WT compared to 7dG, I see no band in 7dG, however in DHX9 nearly the same ratio is blotted but the band intensity looks identical in G4 and controls? LARP1, which was identified previously as a G4 binder here shows robust binding to 7dG as well as the mutG4? KSRP should be a control for a protein binding to unfolded G4s, but also here the calculations of the ratio is strange, nearly the same ratios are written as for the hnRNP, but band intensities vary.

In **Fig. 1E** (submitted version), we placed the ratio values between WT/7dG from the RP-MS next to the western-blot. Since this has been confusing, and as indicated in § 1.9, we deleted these values and added the quantification of the western-blot to show that the differential RNA-protein binding interactions were significant.

2.2.d Most importantly they draw the conclusion that because the hnRNP band is stronger for the 7dG that it binds better to 7dG than to G4. Subsequent in vitro experiments are needed to support this conclusion. Meaning CD spectra of G3A2 constructs as well as binding analysis (e.g. MST, EMSA) are essential to state such a strong claim

As suggested by the Reviewers, we now present data using CD spectra showing that only the G3A2 WT was able to structure into RG4s (see also §1.7). To address the question of the specificity of the interaction, we validated the RP-MS data using both RNA-pull down and reverse IP (**Suppl. Fig. S3B,C**, revised manuscript).

As suggested, we tried EMSA with recombinant H and F proteins but the gel shift was not effective (smeared) probably due to folding issues linked to the denaturation and renaturation steps necessary to recover the two proteins from the inclusion bodies. To further address this point, we performed surface plasmon resonance experiments (as we recently did in (25)) with cytoplasmic extracts lysates, the G3A2 WT/deaza mRNA-coated sensor chips and hnRNP H/F antibody. The results presented in **Suppl. Fig. S3D** (revised manuscript) show the formation of mRNA-ribonucleoprotein complexes with both RNAs but those formed with the G3A2 7dG preferentially associated with hnRNP H/F.

2.2.e Fig. 1F the experimental set up is not clearly stated. in the remaining manuscript the authors use cPDS, why did they perform these experiments with PDS?

cPDS has only recently become commercialized; therefore, some initial experiments were carried out with PDS. We now performed all the experiments with cPDS and kept the experiments with PDS as additional data (supplementary section). Importantly, we obtained similar results with the two ligands (e.g. compare results in **Fig. 1D** and **Suppl. Fig. S3A**), strengthening our conclusion of the preferential interaction of hnRNP H/F with the unfolded RG4s. The experimental set up has been clarified, as requested.

2.2.f Fig. 1F :The choice of hnRNP I as a control is not clear, it is mainly nuclear and why does it bind at all because the pyrimidine do not alter in the constructs and there are no poly pyrimidines int the constructs? Mainly a reverse IP is essential to verify this interaction. Pull down of the proteins and detection of the G3A2.

We are grateful to the Reviewer for raising the question on how to control our affinity chromatography experiments (**Fig. 1D** and **Suppl. Fig. S3A**, submitted version), as it was not mentioned in the first submission. This control is critical because it ensures that the differential RNA-protein interaction effects are solely ascribed to the sequence/structure of interest and not to a different amount of input/bound RNAs or to a possible effect of a specific treatment (cPDS/PDS) on the general protein's ability to associate with RNA. To take into account this issues, we cloned the G3A2 sequence in a plasmid and added flanking nucleotides to 1) have Us in the sequence so that the RNAs can be transcribed with biotinylated Us, and 2) to add a cleavage site (NheI) at the end of the sequence for run-off transcription. We chose the flanking sequence by minimizing the predicted structure stability and creating a binding site for hnRNP I to add an internal control that we previously used to study RNA-protein interactions with cytoplasmic extracts (24). Thus, we controlled our experiments in two ways: by dosing the amount of bait RNA in the eluates and by testing the amount of hnRNP I associated with the two RNAs (sharing an identical sequence). The text has been modified to describe the hnRNP I control (in the Material and Method section).

Regarding the fact that hnRNP I is nuclear, as mentioned above (§ 2.1.c), it is known to shuttle between the nucleus and cytoplasmic compartment (24). As requested, we validated the RNA-pull down results with the reverse IP (**Suppl. Fig. S3B,C**, revised version). The results, in terms of effects and their magnitude, are comparable to those from the RNA affinity chromatography experiments (**Fig. 1C,D** and **Suppl. Fig. S2D** and **S3A**, revised version). The text has been modified to highlight this result, which strengthen our findings on the differential interaction of hnRNP H/F depending on RG4 structuration.

2.3.a Fig. 2A Nevertheless, their approach to check for the localization in different GBM is valid and makes sense, however why are the authors

not surprised to find it mainly in the nucleus? “But codistribute with proteins associated with active translation (lane 50)” Do they mean cytosol fraction? Why is there no fractionation control for the cytosol fraction like tubulin?

As mentioned above, hnRNP H/F like many other RBPs shuttle between the nucleus and the cytoplasm (e.g. (26-28)) and have distinct functions in the two compartments. We were not surprised to find hnRNP H/F proteins in both compartments since it has been previously reported that moderate to high cytoplasmic expression may occur in cells, depending on both tissues and on the normal/tumoral status (29). For active translation sites, we mean cytosol and microsome-associated endoplasmic reticulum (30). The text has been modified accordingly. As suggested, we have added the fractionation control for the cytosol.

The text has been modified accordingly.

2.3.b Fig. 2B : where are the fractions 1-7? This would give a real idea how much of the total cellular hnRNP H/F is interacting with the ribosome and how much is unbound or in low-molecular complexes. If the puromycin just makes the 80S more abundant, is the shift simply based due to changes in polysome profiling?

As the resuspension of fractions 1-6 after precipitation with isopropanol was incomplete, we loaded starting from the 7th fraction. Therefore, we quantified the fraction of hnRNP H/F loaded on polysomes by calculating the ratio between the amount of hnRNP H/F in polysomes and the input. We also analyzed this ratio for eIF4A and DHX36. We found a ratio of 5% for hnRNP H/F and, similarly to Sauer et al. (31), 9,5% for DHX36 and 7% for eIF4A, suggesting a possible role of hnRNP H/F in translation initiation.

Moreover, in **Fig. 2B**, given that mRNPs assembled on untranslated mRNAs may co-sediment in polysomal fractions, we used puromycin, which causes ribosomes dissociation. Unlike puromycin, EDTA and RNAses, two common disrupters, cause complete dissociation of both polysomes and mRNPs, resulting in the total collapse of absorbance in these fractions. As expected with puromycin, we observed both the increase in the 80S and a decrease in the signal in light and heavy polysomal fractions that we quantified as 50% inhibition of ribosomes association. These changes being associated with a shift of hnRNP H/F and other factors, including translation initiation regulators, towards lighter fractions, we concluded that the observed modifications in co-sedimentation were the result of ribosomes dissociation.

We modified the text to stress this point

2.3.c C why is PDS and not cPDS used? 60S and 80S fractions are stronger, shift towards these regions of proteins might be due to G4 but also could be simply be due to indirect effects. PDS alters transcription and

translation at many levels and observed effects are overinterpreted. Lines of the UV profile are blurred compared to other shown polysome profiles

In the revised version, all figures with PDS have been repeated using with cPDS (see also our answer in § 2.2e), including **Fig. 2C** (U251 cell line) and **Suppl. S5B** (U87 cell line) in the revised manuscript, for which the profiles are now sharper. We have chosen cPDS since it was previously reported that the “RNA G-quadruplex-specific ligand, carboxyPDS, does not stabilize DNA G-quadruplexes in nuclei but only traps RNA G-quadruplex structures in the cytoplasm »(15). Importantly, to prevent indirect transcriptional effects, we performed short cell treatments (1-2 hour) with cPDS. These notions have been added to the revised version. See also § 2.6 for further validation using USP1 RNA reporters.

2.4.a Figure 3: the aim here was to re-analyze published datasets and set them into context of new hypothesis. Directly: are hnRNP H/F binding sites G4 regions. For this they used predicted G4 by QGRS mapper, why did the author not use experimental identified G4 regions (Kwok et al., 2016)? lane 285.....bound an important fraction of predicted RG4.. here a more scientific number is essential to validate these analyses. Is this more than expected by chance? What are random control regions? Are the UTRs meant by this? It is not too striking that more is the UTRs, because they are more G-rich than the CDS.

We first employed predicted G4 sites in order to be more inclusive in defining the potential overlap of H/F sites with G4s. We also have performed the same analysis with the dataset by Kwok et al., as wisely suggested by the reviewer. There, observed G4s are markedly more abundant in the 3'UTR (43%) and in the CDS (35%) rather than in 5'UTRs (19.7%), (**Suppl. Figure S7B**). Nevertheless, we replicated Figure 3B (i and ii) and obtained similar results (**Suppl. Figure S7B**, revised version), although the magnitudes of the enrichment are different and reflect the shifted abundance of RG4s in the different regions of the mRNA.

Concerning the fraction of bound RG4s, we have added percentages in the main text, for which the statistical significance of enrichment is shown in **Fig. 3B ii** for the different mRNA regions. As indicated in the Materials and Methods section, the control we used for regions bound by H/F are ten thousand size-matched random regions, effectively controlling for background presence of RG4-like sequences in unbound sites. Concerning the expected abundance of RG4s in the UTRs, GC content is indeed higher at the transcriptome-wide level in the 5'UTR than in the CDS (57.5% vs 52.2%) but is lower in the 3'UTR (44.2%), making the enrichment in the UTRs less granted than could be expected.

2.4.b Figure 3 C GO term analysis is a good idea but analysis needs to be validated experimentally. I agree with the authors that there is a link to DDR but also to many other pathways, the link to GBM is weak and C belongs to the supplement figures.

Our functional analysis indeed showed several pathways. In this manuscript, we focused on validating the impact of hnRNP H/F in the response to DNA damage (**Fig. 5,6**). We believe that the investigation of the role of hnRNP H/F in other pathways is out of scope in this manuscript and would make the main message less straightforward. As regards to the link of hnRNP H/F-bound mRNAs with GBM, we further investigated our results with gene-disease association data collected from the OMIM database and found a significant association with GBM (adjusted p-value=.03284 and .001729 for H and F targets respectively) but also other cancer diseases, including breast cancer (adjusted p-value 0.033 and 1.2E-06), ovarian cancer (adjusted p-value 0.013 and 1.8E-05) etc. These findings support the view (argued in § 1.4) that hnRNP H/F could impact the development and treatment of several cancer types. The discussion has been modified to include these notions. **Fig. 3C** (first submission) was moved to Suppl. Data.

2.4.c Fig. 3D the experiments are well designed, but the conclusion raised by the authors is too strong. First the observed effects are super minor and can be simply be due to experimental variations? Where is the error bar and statistical tests? However, if there are minor changes, these can be RG4 independent and experimental proof are essential to support their strong conclusion.

The translational effects analyzed with the polysome profiling technique are statistically significant (**Fig. 3D,E**, revised manuscript) and can be assessed by both pooling the fractions and analyzing them individually (with replicates in the revised manuscript, **Suppl. Fig. S8B**). To strengthen this result, we used USP1 RNA reporters (**Fig. 3F** and **Suppl. Fig. S8D**, revised manuscript) that showed RG4-dependence but also provided an explanation for the slight but significant modifications in translational efficiency following loss of hnRNP H/F *in cellulo* (this issue has been discussed in § 2.0).

In the revised manuscript, the RG4 dependency is supported by the observation that hnRNP H/F targets were immunoprecipitated by the BG4 antibody (**Fig. 3C**) and their translation was modulated by cPDS (**Fig. 3E**) (with effects similar to those observed after loss of hnRNP H/F (**Fig. 3D**)), and that the expression of USP1 reporters were sensitive to both hnRNP H/F and cPDS treatment but not to RG4 mutations (**Fig. 3F**, **Fig. S8D**).

2.5 Figure 4: formatting errors in C. Why are there not statistical validations in all other panels? Western blots of C and D should be moved to supplement

The revised version contains statistical validation of all panels, and both **Fig. 4C(i)** and **Fig. 4D(ii)** (first submission) were moved in the supplementary section, as suggested. Formatting errors in **Fig. 4C** were also corrected, as requested.

2.6 Fig. 5: The experiments are valid, but the conclusion drawn from then is too strong. That G4 stabilization induces DNA damage is known and anticipated, an addition of ligands changes transcription and translation

at multiple regions. A reporter assay or global analysis combining targets for hnRNP H/F and DHX36 and transcriptional changes in genes associated with DNA repair are needed.

Indeed, it has been established that PDS induces phosphorylation of γ H2AX related to DNA damage (32). For these experiments, we used cPDS, which does not target DNA but cytoplasmic RNA (15). As far as we know, cPDS was not reported to induce the appearance of γ H2AX foci.

Being aware of the indirect effects with cPDS (also see § 2.3c), in order to address the question of the functional effects of cPDS *in cellulo*, we treated GBM cells with this ligand for short time periods (1-2h), so that to exclude transcriptional effects (occurring after several hours). However, to underpin the conclusion that the observed effects are primarily translational and provide further validation of the translational role of the RG4s interacting with hnRNP H/F, we performed polysome profiling in the presence of cPDS and analyzed both translational and transcriptional effects on hnRNP H/F targets. As shown in **Fig. 3E** (revised manuscript), cPDS increased the translation of MECP2 and PRR5 and decreased the translation of VEGF, USP1, CCNA2 and BABAM1. Remarkably, these translational effects were consistent with those observed after hnRNP H/F depletion (**Fig. 3D**, revised manuscript), suggesting that both cPDS treatment and hnRNP H/F depletion resulted in RG4 structuration of the hnRNP H/F targets with consequent (positive or negative) impact on protein synthesis. However, for all (with the exception of the USP1 mRNA, for which the treatment increases the amount of mRNA (i.e. opposite to the translational effect)), these effects were not accompanied by changes in mRNA accumulation (**Suppl. Fig. S8C**, revised manuscript), thus supporting direct translational effects. As suggested by the Reviewer, to further address this issue, we used mRNA reporters. We focused on the USP1 mRNA because this transcript showed a high sensitivity to depletion of hnRNP H/F or DHX36 (**Fig. 3D**) and cPDS treatment (in terms of RNA structuration (**Fig. 3C**) and translational impact (**Fig. 3E**)) but also is strongly immunoprecipitated by both hnRNP H/F and DHX36 (**Fig. 4B**). At first, since cPDS induced a slight but significant modification in mRNA accumulation (**Suppl. Fig. S8C**, revised manuscript), we verified that cPDS had an effect on translational regulation of this target by using *in vitro* transcribed RNAs and *in vitro* translation assays. We observed that cPDS reduced the translation of the reporter capable of RG4 structuration (USP1 WT RNA) but not its modified version unable to form RG4s (USP1 7dG RNA) (**Suppl. Fig. S8D**), indicating that the RG4 in USP1 transcript inhibited mRNA translation. Then, to provide further *in cellulo* confirmation of the direct RG4-dependent translational effect, we tested the effect of cPDS in parallel with PhenDC3, another ligand targeting RG4s and their function in translation (33). We were unable to perform this analysis with cPDS because the treatment induced an alteration in the expression of the control FLuc reporter (data not shown). However, in agreement with the effect of a RG4 ligand on USP1 protein expression (**Fig. 6D**), treatment with PhenDC3 resulted in significant inhibition of the USP1 WT, not the USP1 7dG, expression, indicating that RG4 stabilization repressed USP1 mRNA translation (**Suppl. Fig. S8D**, revised manuscript). This conclusion is further corroborated by experiments with hnRNP H/F silencing, followed by CHX (**Fig.**

5E) or USP1 reporter transfection (**Fig. 3F** and **Suppl. Fig. S8D**, revised manuscript) (discussed in § 2.0). Overall, while we cannot exclude additional indirect effects, these results support the conclusion that hnRNP H/F-RG4 interaction-mediated translational regulation impacts on the DDR. The text has been modified to describe these new results and discuss them.

2.7 The paper and especially the discussion would benefit if the Wolfe et al paper would be included

In the revised version of the manuscript, we have considered Wolfe's work on eIF4A and RG4s in the introduction and in the discussion section.

Minor

- **Lane 148: the term RG4 regions is misleading. Proteins unfold or unwind RG4 but not the regions.**

We modified the sentence, as suggested.

- **Major language and formatting changes are required e.g. lane 166 (to to), lane 859**

We carefully verified language and formatting.

- **Problems in labeling of figures and Supps. E.g. lane 188-189, which is not Supp, it should be 1e. Figure 4D is not mentioned in the main text**

We modified the text, as requested.

- **KSRP named in text, KhSRP in the table, make uniform**

We made uniform the protein names, as indicated.

- **In methods its odd to change in one sentence between two hundred micrograms and 400 hundred micrograms (lane 753), in the later case the “hundred” is also redundant**

We made the requested changes.

- **Western blot procedure is missing, only names of antibodies not sufficient**

The revised manuscript contains a detailed description of the western blot procedure.

+++++

Reviewer 3

We thank Reviewer 3 for the extremely positive assessment of our work and for having emphasized that it represents important advances in the RG4 and translation fields linked to cancer.

+++++

Reviewer 4

We thank the referee for the positive comment highlighting the relevance of our study in the RG4 field. In the verified version, our results are supported by quantitative analysis of the data, thus strengthening our conclusions on role of hnRNP H/F-RG4 interactions in GBM.

4.1.a. the proteomics data is unclear. The authors date that they fine 328 RG4BPs and 42 G-rich BPs. In the supplementary table all proteins are present in all enrichments, irrespective of the RNA input. Firstly the number of interacting proteins seems high, especially for RG4s, and secondly I would have expected some unique interactors that would manifest themselves as present in only one of the conditions. How was data imputation performed on missing values and was this the reason for the lack of presence/absences?

The reviewer correctly assumes that data imputation has shifted the “presence/absence” possibility into the high absolute ratios category.

In such experiments as LC-MS, the measurement is not exhaustive. This means that a no data may either mean “too few to be detected over background” or “too many eluted peptide to process them all”. This is due to the sampling or scanning frequency of the MS and the number of co-eluting peptides. In order to clarify the concerns raised by the Reviewers 1 and 4, we provided a table where imputed values are highlighted in blue. We added two columns (S and T) showing the occurrences of valid values before imputation for each group and which protein corresponds to unique interactors (either 0 valid values in WT and 4 valid values in 7dG or 4 valid values in WT and 0 valid values in 7dG).

In order to explain data imputation, we added the following sentence in the Material and Methods section: “The imputation step allows ratio calculation for all eligible hit (i.e. at least 3 valid values in at least one group), where initially there could have been insufficient data in one condition but enough data in the other condition. We imputed missing data using a random value comprised in the lowest range of LFQ intensities obtained in maxquant”.

4.1.b. Also the authors have chosen a cut-off of 1.5 fold –how is this justified? Are enough replicates run to justify this threshold?

The cut off of 1.5 has been chosen arbitrary. With four number of replicates the calculated q-value after the t-tests were satisfactory (column Y) for the selected protein list.

4.1.c. Of the new RBPs identified, how many of them have been identified in three recently published papers that use new approaches to expand the RBP databases (PTEX1, XRNAX 2 and OOPS 3 papers)?

We thank the Reviewer for underlining this point since our analysis based on a single work was too restrictive, thus skewing our analysis and discussion on our RP-MS results. As suggested, we intersected our RP-MS data with the recent interactome capture method (**Suppl. Table S2**, revised version). For the Venn diagram, proteins found in at least 2 of these works were indicated as RBPs.

This point was discussed also in § 2.1.b. The text has been modified.

4.1.d. The proteomics data seem incomplete, and I would have liked to have seen data for proteins binding to the mutated sequence, UCUGCGAAGUGAAGUGAAGCGAUC, to give more credence to the rest of the proteomics data and filter further background interactions not specific for the structures under consideration.

The referee raised an important point that we have considered when designing the RP-MS. To answer the question of what factors bind RG4 or G-rich we have chosen an RP-MS approach with 4 replicates to obtain quantitative data, with a high level of sensitivity, excellent reproducibility. We reasoned that both the G3AA WT and 7dG RNAs would have captured proteins interacting aspecifically and since we are interested in differential interactors, this would have been excluded from the analysis. In the ideal setup, several controls are needed: a control without G-stretches to verify for the G-requirement, a control for proteins that aspecifically recognize RNA structures (e.g. stem-loop, other), a control to exclude RG4 conformation-specific (parallel or antiparallel (34)) or to study quartet-number specific binding. All these controls are equally important, but their implementation would have changed the effort and scope of the study, in which the identification of RG4-RBP was a first step before investigating the function and physiopathological consequences. We have chosen an experimental set up that provides an overview of differential interactions, and opens up interesting and new perspectives, for example on proteins that are not known to function with RG4s, such as LARP1 or the m6A-binding factors.

4.2.a the authors cherry picked proteins of interest from their proteomics data with no clear reasons why other in the list should be ignored and then carry out pulldowns and westerns.

In the revised manuscript, we indicated that the proteins tested are those that are often found in RP-MS data (hnRNP H/F, DHX36, DHX9, DDX3X) while others remain (LARP1, KHSRP) to be characterized in terms of their function associated with RG4s.

4.2.b The data presented in figure 1E are not compelling. The statement in the test (line 193) saying that the pulldown of the helicases showed 'less' interaction G3A2 7dG than with G3A2 WT is not satisfactory. Where are the replicates and error bars that support this statement?

To reinforce the results in this figure, we have quantified the data that now show significant differences in binding and corroborate the RP-MS data. Following the Reviewers' advices, this preferential interaction was further demonstrated in the revised manuscript by 1) pulling down the RNA by immunoprecipitating RG4/G-rich binding proteins (**Suppl. Fig. S3B,C**), 2) by surface plasmon resonance (**Suppl. Fig. S3D**) and further characterized by 3) RNA affinity chromatography in the presence of cPDS (**Fig. 1D**) (in addition to PDS).

3. all the conclusions made about the data portrayed in figured 2 would be much better supported by the use of replicates. In particular, the western blot displayed in figure 2C is of poor quality, and based on it I do not accept the conclusions made.

To address this point which was also raised by the reviewer (§ 2.4.c), together with the comment on the fact of homogenizing the experiments with a single ligand (cPDS) (§ 2.2.e), we repeated the experiments in **Fig. 2C** with the cPDS and validated them in a second cell line (U87) (**Suppl. Fig. S5B**). Replicates of **Fig. 2C** have been quantified to strengthen our conclusion on the ability of cPDS to modify the association of hnRNP H/F with polysomes. We observed similar effects with cPDS and PDS.

4. In general all polysomal profiling data is badly represented. In figure 3 (and figure 6) the lack of replicates make the subtle changes in distribution of proteins among the different populations of monosomes and polysomes difficult to accept.

The results of the polysomes are now based on a pool analysis with significant effects (**Fig. 3D** and **6B**, revised version) and on the individual fractions (with replicates, **Suppl. Fig. S8** and **S11**, revised version). The translational effects were further supported by polysomal analyses in the presence of cPDS (**Fig. 3E**, revised version) and with reporters in two cell lines (**Fig. 3F** and **Suppl. Fig. S8D**, revised version). We believe that all these evidences support the translational effects that were reproducible and significant regardless of the GBM cell line, the specific treatment (cPDS/PDS) or methodology to analyze them (polysome profiling, reporter assays).

5. It is not clear to me why the immunoprecipitations described figure 4 were not carried out in the C and M fraction separately, as interaction of hnRNP H/F with the translational machinery may be different in the cytosol compared with the ER.

We agree with the reviewer that we could have pushed forward our analysis to define in which compartment (cytoplasm or RE) the interactions take place. However, the aim here was to provide a first evidence of the translational role of hnRNP H/F, which remains poorly explored to date. For this, we have performed classical polysome experiments based on hypotonic lysis, followed by a treatment with a mild detergent to solubilize cytosolic and ER-associated ribosomes, as described in (35). Thereafter, our aim was to describe the underlying mechanism under the same conditions used to investigate the function of hnRNP H/F in translation. In future studies, it will be necessary to explore more precisely the C and M fraction separately. We referred to this intriguing possibility in the discussion.

6. **I do not agree with the statement from line 355 onwards about the preferences for hnRNP H/F and DHX36 for folded or unfolded RG4 and the dependency on DHX36 by hnRNP H/F for binding to its targets. The data on which this statement is made is mainly qualitative (i.e. pictures of westerns) that without replication, makes the statement extremely speculative.**

In the revised version, the differential interaction depending on the G3A2 folding is supported by pull-down (with replicates and quantifications) using either RNAs (**Fig. 1C,D**) or proteins as bait (**Suppl. Fig. S3B,C**). Regarding hnRNP H/F targets, **Fig. 4** now shows significant effects for the interactions of the two proteins with the same targets (**Fig. 4B**) as well as for their interaction with mRNA targets in the absence of the protein partner (**Fig. 4C,D**).

REFERENCES

1. Decorsiere A, Cayrel A, Vagner S, Millevoi S. Essential role for the interaction between hnRNP H/F and a G quadruplex in maintaining p53 pre-mRNA 3'-end processing and function during DNA damage. *Genes Dev.* 2011;25(3):220-5.
2. Guo JU, Bartel DP. RNA G-quadruplexes are globally unfolded in eukaryotic cells and depleted in bacteria. *Science.* 2016;353(6306).
3. Yang SY, Lejault P, Chevrier S, Boidot R, Robertson AG, Wong JMY, et al. Transcriptome-wide identification of transient RNA G-quadruplexes in human cells. *Nat Commun.* 2018;9(1):4730.
4. Thandapani P, Song J, Gandin V, Cai Y, Rouleau SG, Garant JM, et al. Aven recognition of RNA G-quadruplexes regulates translation of the mixed lineage leukemia protooncogenes. *Elife.* 2015;4.
5. Haeusler AR, Donnelly CJ, Periz G, Simko EA, Shaw PG, Kim MS, et al. C9orf72 nucleotide repeat structures initiate molecular cascades of disease. *Nature.* 2014;507(7491):195-200.
6. McRae EKS, Booy EP, Moya-Torres A, Ezzati P, Stetefeld J, McKenna SA. Human DDX21 binds and unwinds RNA guanine quadruplexes. *Nucleic Acids Res.* 2017;45(11):6656-68.

7. Serikawa T, Spanos C, von Hacht A, Budisa N, Rappsilber J, Kurreck J. Comprehensive identification of proteins binding to RNA G-quadruplex motifs in the 5' UTR of tumor-associated mRNAs. *Biochimie*. 2018;144:169-84.
8. Herdy B, Mayer C, Varshney D, Marsico G, Murat P, Taylor C, et al. Analysis of NRAS RNA G-quadruplex binding proteins reveals DDX3X as a novel interactor of cellular G-quadruplex containing transcripts. *Nucleic Acids Res*. 2018;46(21):11592-604.
9. Castello A, Fischer B, Eichelbaum K, Horos R, Beckmann BM, Strein C, et al. Insights into RNA biology from an atlas of mammalian mRNA-binding proteins. *Cell*. 2012;149(6):1393-406.
10. Baltz AG, Munschauer M, Schwanhaussner B, Vasile A, Murakawa Y, Schueler M, et al. The mRNA-bound proteome and its global occupancy profile on protein-coding transcripts. *Mol Cell*. 2012;46(5):674-90.
11. Yamazaki T, Liu L, Lazarev D, Al-Zain A, Fomin V, Yeung PL, et al. Corrigendum: TCF3 alternative splicing controlled by hnRNP H/F regulates E-cadherin expression and hESC pluripotency. *Genes Dev*. 2019;33(11-12):733-6.
12. Mauger DM, Lin C, Garcia-Blanco MA. hnRNP H and hnRNP F complex with Fox2 to silence fibroblast growth factor receptor 2 exon IIIc. *Mol Cell Biol*. 2008;28(17):5403-19.
13. Brannan KW, Yeo GW. From Protein-RNA Predictions toward a Peptide-RNA Code. *Mol Cell*. 2016;64(3):437-8.
14. Kechavarzi B, Janga SC. Dissecting the expression landscape of RNA-binding proteins in human cancers. *Genome Biol*. 2014;15(1):R14.
15. Biffi G, Di Antonio M, Tannahill D, Balasubramanian S. Visualization and selective chemical targeting of RNA G-quadruplex structures in the cytoplasm of human cells. *Nat Chem*. 2014;6(1):75-80.
16. Urdaneta EC, Vieira-Vieira CH, Hick T, Wessels HH, Figini D, Moschall R, et al. Purification of cross-linked RNA-protein complexes by phenol-toluol extraction. *Nat Commun*. 2019;10(1):990.
17. Trendel J, Schwarzl T, Horos R, Prakash A, Bateman A, Hentze MW, et al. The Human RNA-Binding Proteome and Its Dynamics during Translational Arrest. *Cell*. 2019;176(1-2):391-403 e19.
18. Perez-Perri JI, Rogell B, Schwarzl T, Stein F, Zhou Y, Rettel M, et al. Discovery of RNA-binding proteins and characterization of their dynamic responses by enhanced RNA interactome capture. *Nat Commun*. 2018;9(1):4408.
19. Queiroz RML, Smith T, Villanueva E, Marti-Solano M, Monti M, Pizzinga M, et al. Comprehensive identification of RNA-protein interactions in any organism using orthogonal organic phase separation (OOPS). *Nat Biotechnol*. 2019;37(2):169-78.
20. Gerstberger S, Hafner M, Tuschl T. A census of human RNA-binding proteins. *Nat Rev Genet*. 2014;15(12):829-45.
21. Gerstberger S, Hafner M, Ascano M, Tuschl T. Evolutionary conservation and expression of human RNA-binding proteins and their role in human genetic disease. *Adv Exp Med Biol*. 2014;825:1-55.
22. Michael WM, Siomi H, Choi M, Pinol-Roma S, Nakielnny S, Liu Q, et al. Signal sequences that target nuclear import and nuclear export of pre-mRNA-binding proteins. *Cold Spring Harb Symp Quant Biol*. 1995;60:663-8.
23. Cho S, Kim JH, Back SH, Jang SK. Polypyrimidine tract-binding protein enhances the internal ribosomal entry site-dependent translation of p27Kip1 mRNA and modulates transition from G1 to S phase. *Mol Cell Biol*. 2005;25(4):1283-97.

24. Lamaa A, Le Bras M, Skuli N, Britton S, Frit P, Calsou P, et al. A novel cytoprotective function for the DNA repair protein Ku in regulating p53 mRNA translation and function. *EMBO Rep.* 2016;17(4):508-18.
25. Franchini DM, Lanvin O, Tosolini M, Patras de Campaigno E, Cammas A, Pericart S, et al. Microtubule-Driven Stress Granule Dynamics Regulate Inhibitory Immune Checkpoint Expression in T Cells. *Cell Rep.* 2019;26(1):94-107 e7.
26. Bava FA, Eliscovich C, Ferreira PG, Minana B, Ben-Dov C, Guigo R, et al. CPEB1 coordinates alternative 3'-UTR formation with translational regulation. *Nature.* 2013;495(7439):121-5.
27. Caceres JF, Sreaton GR, Krainer AR. A specific subset of SR proteins shuttles continuously between the nucleus and the cytoplasm. *Genes Dev.* 1998;12(1):55-66.
28. Van Dusen CM, Yee L, McNally LM, McNally MT. A glycine-rich domain of hnRNP H/F promotes nucleocytoplasmic shuttling and nuclear import through an interaction with transportin 1. *Mol Cell Biol.* 2010;30(10):2552-62.
29. Honore B, Baandrup U, Vorum H. Heterogeneous nuclear ribonucleoproteins F and H/H' show differential expression in normal and selected cancer tissues. *Exp Cell Res.* 2004;294(1):199-209.
30. Reid DW, Nicchitta CV. Diversity and selectivity in mRNA translation on the endoplasmic reticulum. *Nat Rev Mol Cell Biol.* 2015;16(4):221-31.
31. Sauer M, Juranek SA, Marks J, De Magis A, Kazemier HG, Hilbig D, et al. DHX36 prevents the accumulation of translationally inactive mRNAs with G4-structures in untranslated regions. *Nat Commun.* 2019;10(1):2421.
32. Rodriguez R, Miller KM, Forment JV, Bradshaw CR, Nikan M, Britton S, et al. Small-molecule-induced DNA damage identifies alternative DNA structures in human genes. *Nat Chem Biol.* 2012;8(3):301-10.
33. Cammas A, Dubrac A, Morel B, Lamaa A, Touriol C, Teulade-Fichou MP, et al. Stabilization of the G-quadruplex at the VEGF IRES represses cap-independent translation. *RNA Biol.* 2015;12(3):320-9.
34. Xiao CD, Ishizuka T, Xu Y. Antiparallel RNA G-quadruplex Formed by Human Telomere RNA Containing 8-Bromoguanosine. *Sci Rep.* 2017;7(1):6695.
35. Gandin V, Sikstrom K, Alain T, Morita M, McLaughlan S, Larsson O, et al. Polysome fractionation and analysis of mammalian translatoemes on a genome-wide scale. *J Vis Exp.* 2014(87).

Reviewers' comments:

Reviewer #1 (Remarks to the Author):

After review, this paper is now of great quality to be published in Nature comm.
Nice job!

Reviewer #2 (Remarks to the Author):

NCOMMS-19-11272A : hnRNP H/F drive RNA G-quadruplex-mediated translation linked to genomic instability and therapy resistance in glioblastoma
By Herviou, LeBras et al.

Reviewer 2

The authors have performed a series of experiments that have significantly improved the manuscript. In the presented data support better the model, (Figure 7) but there are still remaining open questions mainly regarding the connection to DDR.

2.1.b

The presented venn-diagram is better, but it is not easy to understand. Also the authors presented LARP1 as a novel G4 binder. This protein has been identified before in Vlasenok et al (<https://doi.org/10.1016/j.dib.2018.02.081>)

In general, their list of proteins should be compared to this paper and this paper needs to be cited and discussed.

Currently the most prominent example of a RBP and G4 function in cancer is Wolfe et al Nature. In this version of the manuscript at least they cite this work but the overlap of targets, function etc is not discussed

2.3.b

Their argument is OK, but it is possible to analyze and precipitate fraction 1-6 with a modified polysome protocol. With the here presented data it looks like it that DHX36 is more enriched at polysomes. The comparison, they have performed to Sauer et al is very useful and supports their data (in the rebuttal). But it would be good to name their numbers and discuss this paper in the main paper. Furthermore, how many targets overlap between hnRNAP and DHX36? Do they overlap only in the 3' or 5' UTR? Did the authors compare their data with the recent paper of the Balasubramanien group? How showed a different polysome gradient for DHX36?

In general, they start with asking the question if only those targets which harbor a G4 are altered in translation. They see an overlap to G4s (11%) but this does not address this question. Pulse SILAC or polysome gradients are needed to address this question.

2.4.b

This part is still very weak and seems constructed.

Are real expression (not on a reporter) or translation of repair proteins affected upon silencing hnRNAP H/F changed? Y-H2Ax signal are not a valid DDR. It is very error prone. Rad51 foci would be better. Meaning do more G4 form in the cytoplasm after silencing of hnRNAP how is this affecting DRR? Is this due to targets? How many proteins of the DDR are targeted by both DHX36 and hnRNAP?

More specific comments

Lane 170: " we characterized 328 RG4-BP.." how did they characterize those proteins?

Experimentally

Lane 210-214: overstatement of data. This is not the first demonstration of RBPs binding only to G-rich. Previous studies that used mutated G4s also had a G-rich control. Please check remaining manuscript not to overstate experimental conclusion. Avoid discussion in the results section

Lane 253: expression levels where compared to?

Lane 254: over-interpretation of the data

Lane 296, the word however is wrong here. Because not translational changes are observed it is expected to see no apoptosis. Better would be here to state if the cells after silencing have a phenotype

Lane 305 & 434 word strikingly should not be used in the results. Most importantly the hear mentioned results are good but not strikingly

Lane 312: they state widespread of G4s, but identified only 11% overlap? Why so little? What are the other targets? Are they more important?

Lane 335, which positive and negative control did they use for this experiment?

Lane 486 -495 the hear presented data are weak to argue for a joint function and a relevance for GBM. Could they test if the cells grow slower and migrate less after silencing of hnRNAP?

Reviewer #4 (Remarks to the Author):

The authors have submitted a much improved manuscript that goes a long way to address my initial concerns.

I have a few points (in green) that still remain and require clarification.

4.1.a. the proteomics data is unclear. The authors date that they fine 328 RG4BPs and 42 G-rich BPs. In the supplementary table all proteins are present in all enrichments, irrespective of the RNA input. Firstly the number of interacting proteins seems high, especially for RG4s, and secondly I would have expected some unique interactors that would manifest themselves as present in only one of the conditions. How was data imputation performed on missing values and was this the reason for the lack of presence/absences?

The reviewer correctly assumes that data imputation has shifted the “presence/absence” possibility into the high absolute ratios category.

In such experiments as LC-MS, the measurement is not exhaustive. This means that a no data may either mean “too few to be detected over background” or “too many eluted peptide to process them all”. This is due to the sampling or scanning frequency of the MS and the number of co-eluting peptides. In order to clarify the concerns raised by the Reviewers 1 and 4, we provided a table where imputed values are highlighted in blue. We added two columns (S and T) showing the occurrences of valid values before imputation for each group and which protein corresponds to unique interactors (either 0 valid values in WT and 4 valid values in 7dG or 4 valid values in WT and 0 valid values in 7dG).

In order to explain data imputation, we added the following sentence in the Material and Methods section: “The imputation step allows ratio calculation for all eligible hit (i.e. at least 3 valid values in at least one group), where initially there could have been insufficient data in one condition but enough data in the other condition. We imputed missing data using a random value comprised in the lowest range of LFQ intensities obtained in maxquant”.

I am confused by the above. In table S1, the blue entries are imputed, but the values seem very high. Was the random value imputed as a function of MaxQuant software –why was this method chosen?

The description on the front page of table S1 ‘These are the IDs of those proteins that have at least half of the peptides that the leading protein has’ is very confusing – please clarify.

Also it would be good on the front page to describe what the blue and red backgrounds refer to on the second page

4.1.b. Also the authors have chosen a cut-off of 1.5 fold –how is this justified?

Are enough replicates run to justify this threshold?

The cut off of 1.5 has been chosen arbitrary. With four number of replicates the calculated q-value after the t-tests were satisfactory (column Y) for the selected protein list.

How were these data corrected for multiple testing – this needs to be added to the methods section with a statement about what constitutes a satisfactory q-value.

This section very much reads as if Perseus was used as a black box by the authors.

4.1.c. Of the new RBPs identified, how many of them have been identified in three recently published papers that use new approaches to expand the RBP databases (PTEX1, XRNAX 2 and OOPS 3 papers)?

We thank the Reviewer for underlining this point since our analysis based on a single work was too restrictive, thus skewing our analysis and discussion on our RP-MS results. As suggested, we intersected our RP-MS data with the recent interactome capture method (**Suppl. Table S2**, revised version). For the Venn diagram, proteins found in at least 2 of these works were indicated as RBPs.

This point was discussed also in § 2.1.b. The text has been modified.

4.1.d. The proteomics data seem incomplete, and I would have liked to have seen data for proteins binding to the mutated sequence, UCGCGAAGUGAAGUGAAGCGAUC, to give more credence to the rest of the proteomics data and filter further background interactions not specific for the structures under consideration.

The referee raised an important point that we have considered when designing the RP-MS. To answer the question of what factors bind RG4 or G-rich we have chosen an RP-MS approach with 4 replicates to obtain quantitative data, with a high level of sensitivity, excellent reproducibility. We reasoned that both the G3AA WT and 7dG RNAs would have captured proteins interacting aspecifically and since we are interested in differential interactors, this would have been excluded from the analysis. In the ideal setup, several controls are needed: a control without G-stretches to verify for the G-requirement, a control for proteins that aspecifically recognize RNA structures (e.g. stem-loop, other), a control to exclude RG4 conformation-specific (parallel or antiparallel (34)) or to study quartet-number specific binding. All these controls are equally important, but their implementation would have changed the effort and scope of the study, in which the identification of RG4-RBP was a first step before investigating the function and physiopathological consequences. We have chosen an experimental set up that provides an overview of differential interactions, and opens up interesting and new perspectives, for example on proteins that are not known to function with RG4s, such as LARP1 or the m6A-binding factors.

I am not convinced by the above argument. The authors are saying here that with four replicates any differences between G3AAWT and 7dG are most important and the other controls, although nice to have, are not necessary to determine the difference in these binding partners, but surely the controls are needed to determine background that might be different for the G3AAWT and 7dG interactors.

4.2.a the authors cherry picked proteins of interest from their proteomics data with no clear reasons why other in the list should be ignored and then carry out pulldowns and westerns.

In the revised manuscript, we indicated that the proteins tested are those that are often found in RP-MS data (hnRNP H/F, DHX36, DHX9, DDX3X) while others remain (LARP1, KHSRP) to be characterized in terms of their function associated with RG4s.

4.2.b The data presented in figure 1E are not compelling. The statement in the test (line 193) saying that the pulldown of the helicases showed 'less' interaction G3A2 7dG than with G3A2 WT is not satisfactory. Where are the replicates and error bars that support this statement?

To reinforce the results in this figure, we have quantified the data that now show significant differences in binding and corroborate the RP-MS data. Following the Reviewers' advices, this preferential interaction was further demonstrated in the revised manuscript by 1) pulling down the RNA by immunoprecipitating RG4/G-rich binding proteins (**Suppl. Fig. S3B,C**), 2) by surface plasmon resonance (**Suppl. Fig. S3D**) and further characterized by 3) RNA affinity chromatography in the presence of cPDS (**Fig. 1D**) (in addition to PDS).

3. all the conclusions made about the data portrayed in figured 2 would be much better supported by the use of replicates. In particular, the western blot displayed in figure 2C is of poor quality, and based on it I do not accept the conclusions made.

To address this point which was also raised by the reviewer (§ 2.4.c), together with the comment on the fact of homogenizing the experiments with a single ligand (cPDS) (§ 2.2.e), we repeated the experiments in **Fig. 2C** with the cPDS and validated them in a second cell line (U87) (**Suppl. Fig. S5B**). Replicates of **Fig. 2C** have been quantified to strengthen our conclusion on the ability of cPDS to modify the association of hnRNP H/F with polysomes. We observed similar effects with cPDS and PDS.

4. In general all polysomal profiling data is badly represented. In figure 3 (and figure 6) the lack of replicates make the subtle changes in distribution of proteins among the different populations of monosomes

and polysomes difficult to accept.

The results of the polysomes are now based on a pool analysis with significant effects (**Fig. 3D** and **6B**, revised version) and on the individual fractions (with replicates, **Suppl. Fig. S8** and **S11**, revised version). The translational effects were further supported by polysomal analyses in the presence of cPDS (**Fig. 3E**, revised version) and with reporters in two cell lines (**Fig. 3F** and **Suppl. Fig. S8D**, revised version). We believe that all these evidences support the translational effects that were reproducible and significant regardless of the GBM cell line, the specific treatment (cPDS/PDS) or methodology to analyze them (polysome profiling, reporter assays).

- 5. It is not clear to me why the immunoprecipitations described figure 4 were not carried out in the C and M fraction separately, as interaction of hnRNP H/F with the translational machinery may be different in the cytosol compared with the ER.**

We agree with the reviewer that we could have pushed forward our analysis to define in which compartment (cytoplasm or RE) the interactions take place. However, the aim here was to provide a first evidence of the translational role of hnRNP H/F, which remains poorly explored to date. For this, we have performed classical polysome experiments based on hypotonic lysis, followed by a treatment with a mild detergent to solubilize cytosolic and ER-associated ribosomes, as described in (35). Thereafter, our aim was to describe the underlying mechanism under the same conditions used to investigate the function of hnRNP H/F in translation. In future studies, it will be necessary to explore more precisely the C and M fraction separately. We referred to this intriguing possibility in the discussion.

- 6. I do not agree with the statement from line 355 onwards about the preferences for hnRNP H/F and DHX36 for folded or unfolded RG4 and the dependency on DHX36 by hnRNP H/F for binding to its targets. The data on which this statement is made is mainly qualitative (i.e. pictures of westerns) that without replication, makes the statement extremely speculative.**

In the revised version, the differential interaction depending on the G3A2 folding is supported by pull-down (with replicates and quantifications) using either RNAs (**Fig. 1C,D**) or proteins as bait (**Suppl. Fig. S3B,C**). Regarding hnRNP H/F targets, **Fig. 4** now shows significant effects for the interactions of the two proteins with the same targets (**Fig. 4B**) as well as for their interaction with mRNA targets in the absence of the protein partner (**Fig. 4C,D**).

NCOMMS-19-11272A : *hnRNP H/F drive RNA G-quadruplex-mediated translation linked to genomic instability and therapy resistance in glioblastoma*

By Herviou, LeBras et al.

We are grateful to the reviewers for their comments and suggestions that contributed to further expanding and strengthening the study. To improve the figures in the new version of the manuscript, we have added different colors in the histograms, purple for hnRNP H/F, green for DHX36, red and orange for cPDS or PDS treatments, respectively. For clarity sake, the modifications in the main text added in this second revision are indicated in blue (in red from the first revision).

Detailed Response to Reviewers' Comments:

Reviewer #1

**After review, this paper is now of great quality to be published in Nature comm.
Nice job!**

We thank the reviewer for his comments and enthusiasm.

Reviewer #2

The authors have performed a series of experiments that have significantly improved the manuscript. In the presented data support better the model, (Figure 7) but there are still remaining open questions mainly regarding the connection to DDR.

We are delighted with the reviewer's positive comments regarding the manuscript's improvement during the first review.

#2.1.b The presented venn-diagram is better, but it is not easy to understand.

As suggested, we slightly modified the Venn diagram and improved the figure legend.

Also the authors presented LARP1 as a novel G4 binder. This protein has been identified before in Vlasenok et al (<https://doi.org/10.1016/j.dib.2018.02.081>) In general, their list of proteins should be compared to this paper and this paper needs to be cited and discussed.

We respectfully disagree with the reviewer on this point since the Vlasenok paper focuses on DNA G4s. Additionally, the G4 binding partners were identified not using G4 pull-down coupled to MS (as in all the others studies considered in our study) but by profiling small molecule-protein interactions on human protein microarrays.

Currently the most prominent example of a RBP and G4 function in cancer is Wolfe et al Nature. In this version of the manuscript at least they cite this work but the overlap of targets, function etc is not discussed.

Indeed, Wolfe et al proposed that eIF4A regulates the translation of mRNAs with (CGG)₄ motifs in the 5'UTR. While more recent works revisited this notion (Waldron NAR 2018; Chan Nat Comm 2019), we did not include the eIF4A targets (Wolfe Nature 2014) in our analysis because hnRNP H/F bind patterns with G triplets. We added this notion to the discussion section of the manuscript (first round of revision).

2.3.b Their argument is OK, but it is possible to analyze and precipitate fraction 1-6 with a modified polysome protocol. The comparison, they have performed to Sauer et al is very useful and supports their data (in the rebuttal). But it would be good to name their numbers and discuss this paper in the main paper.

We agree with the reviewer but we tried different protocols and the one we used in the manuscripts works well for the polysomal fractions and allows reproducible results. However, including fractions 1-6 would not change the polysome/input ratio that we found comparable to Sauer et al. As suggested, we have now added the comparison with Sauer's data to the main text.

Furthermore, how many targets overlap between hnRNAP and DHX36? Do they overlap only in the 3' or 5' UTR?

As suggested we intersected the CLIP data from hnRNP H/F and DHX36. We found 327 genes overlapping between H/F and DHX36, with overlaps distributed in the whole mRNA (5'UTR, CDS, and 3'UTR). We have added a supplementary figure describing this overlap to the revised version of the manuscript (**Suppl. Fig. S12**)

Did the authors compare their data with the recent paper of the Balasubramanian group? How showed a different polysome gradient for DHX36?

We agree that the distribution of DHX36 is different in Murat's work (Fig. 3B in Murat Genome Biology 2019) but the S6 profile also differs, with a greater shift towards heavy polysomes compared to our profiles. Noteworthy, our profiles for DHX36 are similar to Sauer et al (Nat Comm. 2019). Different cell lines and protocols could explain profiled that might not be completely equivalent.

In general, they start with asking the question if only those targets which harbor a G4 are altered in translation. They see an overlap to G4s (11%) but this does not address this question. Pulse SILAC or polysome gradients are needed to address this question.

Indeed, in order to answer the question of the function of the interactions between hnRNP H/F and the RG4-forming G-rich sequences, we first explored in cellulo RNA-protein interactions and then studied in depth the effect of these associations on the translation of specific RG4-containing targets involved in the DNA damage repair pathway. Analyzing the full hnRNP H/F translome by performing pulse SILAC or polysome coupled to RNA-seq would answer a different and more general question about the role of hnRNP H/F in the translation of mRNA, which deserves to be addressed exhaustively in a future study. The discussion has been modified accordingly to stress this point.

We would like to point out that the 11% overlap corresponded to the fraction of hnRNP H/F-bound RG4s over the total predicted RG4s. We apologize if this notion was unclear in the text and in the figures. To avoid any misunderstanding on this important point of the manuscript, we have modified Fig. 3Bi (by indicating precisely the Y-axis), the main text and the figure legend.

§ 2.4.b This part is still very weak and seems constructed. Are real expression (not on a reporter) or translation of repair proteins affected upon silencing hnRNAP H/F changed?

We sincerely apologize for having misunderstood the comment from the first revision regarding the evidence supporting the link between hnRNP H/F and DDR. We believe that this link is consistently supported by both high- (enrichment analysis of hnRNP H/F targets containing RG4s) and low-throughput evidences (DDR mRNA target interaction and translatability (**Fig. 3 and 4**); DDR marker signals as function of translation regulation (**Fig. 5**); accumulation of USP1 protein depending on hnRNP H/F, DHX36 and RG4 folding (**Fig. 6**); and the consequences of hnRNP H/F on protein ubiquitination).

Y-H2Ax signal are not a valid DDR. It is very error prone. Rad51 foci would be better.

We chose H2AX phosphorylation as a marker of DDR based on a reference paper in the field (Bonner Nat Rev Cancer 2008) reporting that: i) it is a universal and early marker for DDR created by all three major apical DDR kinases (ii) it is known to be phosphorylated in response to DSBs and other DNA lesions. In agreement, H2AX phosphorylation has been previously used to support the existence of a link between DNA G4s and DDR activation (Rodriguez, Nat Cell Biol 2012; Wang Nat Comm 2019). In addition, both signals from H2AX phosphorylation and 53-BP1 (both quantified in **Fig. 5**) are also used to detect damage in USP1-depleted cells (Ogrunc, Cell Rep 2016). Unlike phosphorylated H2AX, Rad51 foci are formed during homologous recombination repair and are part of downstream signaling pathways.

We added the reference Bonner Nat Rev Cancer 2008 in the main text.

Meaning do more G4 form in the cytoplasm after silencing of hnRNAP, how is this affecting DRR? Is this due to targets?

We thank the referee for raising this important point. In the revised manuscript, **Fig. 4E** and **Suppl. Fig. S9E** now shows that hnRNP H/F depletion increased the BG4 signal similarly to DHX36 silencing or cPDS-mediated RG4 stabilization and in a RNA-dependent manner. Similar results for DHX36 and cPDS were previously observed by Sauer et al (Nat Comm 2019). These new data strengthen our results on the hnRNP H/F-mediated regulation of RG4s *in cellulo* and together with **Fig. 3** showing that hnRNP H/F regulated the translation of DDR-related genes, **Fig. 5** showing that the regulation at the translational level impacts on the accumulation of DDR markers and, finally, **Fig. 6** validating the DDR protein USP1 as a target of the synergistic RG4-resolving mechanism, we strongly believe that the burden of proofs for a link between hnRNP H/F and genome instability via RG4-mediated translational regulation has been met. Taken together, these data show that the hnRNP H/F-DDR link is mediated by the translational targets of hnRNP H/F.

How many proteins of the DDR are targeted by both DHX36 and hnRNAP?

This point has been discussed in **§ 2.3b. Suppl Fig. S12** now shows that DHX36 and hnRNP H/F share 74 common targets involved in the DDR.

More specific comments

Lane 170: “ we characterized 328 RG4-BP..” how did they characterize those proteins? Experimentally

We modified the sentence as suggested.

Lane 210-214: overstatement of data. This is not the first demonstration of RBPs binding only to G-rich. Previous studies that used mutated G4s also had a G-rich

control. Please check remaining manuscript not to overstate experimental conclusion. Avoid discussion in the results section

As far as we know, our manuscript reports for the first time an exhaustive and quantitative identification of the proteins binding the G-rich RNAs. Indeed, in Von Hacht (NAR 2004), in addition to G-mut sequences, they used control sequences with higher guanine content and two G-stretches. However, unlike deaza-containing G-rich sequences, these RNAs may still form inter-molecular RG4s. Moreover, only the two most prominent proteins bound to these control RNAs were identified. In other studies performing RNA pull-down coupled to MS only the control sequences devoid of G-stretches were used (Serikawa Biochimie 2018; Herdy NAR 2019; McRae NAR 2017). On the basis of these studies, we modified the sentence with “and **provided the first comprehensive evidence** of which proteins bind structured RG4s and which ones prefer to bind the G-rich sequence *per se*.”

In the discussion, we verified that the sentence “In contrast to previous RP-MS data sets (^{8, 9, 10, 11, 12}), we were able to capture and identify proteins binding to folded and unfolded RG4s by comparing native and 7dG-substituted G3A2 RNAs” corresponds to the state-of-the-art in the RG4 RBP field.

Lane 253: expression levels where compared to?

Sorry for this error, and thank you for noticing. They were compared to normal brain. We have now completed the sentence.

Lane 254: over-interpretation of the data

The sentence has been replaced by «*We found that hnRNP H/F family members displayed higher expression levels in GBM (Suppl. Fig. S4A), suggesting a potential role for both RBPs in GBM gene expression reprogramming*»

Lane 296, the word however is wrong here. Because not translational changes are observed it is expected to see no apoptosis. Better would be here to state if the cells after silencing have a phenotype

As suggested, we skipped “however” and rephrased the sentence, to read:

Polysomal profile was slightly altered by hnRNP H/F depletion (Fig. 3A, Suppl. Fig. S6C), indicating that cells deficient in hnRNP H/F are not globally defective in protein synthesis. Neither apoptosis nor proliferations were affected under these treatment conditions (Suppl. Fig. S6D,E), suggesting that changes in translation efficiency after hnRNP H/F silencing were not directly related to these processes.

Lane 305 & 434 word strikingly should not be used in the results. Most importantly the hear mentioned results are good but not strikingly. Lane 312: they state widespread of G4s, but identified only 11% overlap? Why so little? What are the other targets? Are they more important?

We respectfully disagree with the reviewer's suggestion to tone down “strikingly” in lane 305. Indeed, the observation that hnRNP H/F could bind such a proportion of RG4 over the transcriptome (discussed above, see § 2.3.b) is indeed surprising and underlines the potential importance of this interaction.

We slightly modified the sentence (underlined) by “Strikingly, hnRNP H/F bound an important

fraction of predicted *RG4s* *over all predicted RG4s in the transcriptome* (11% of 5'UTR, 2.7% of CDS, and 11.4% of 3'UTR) (Fig. 3B and Suppl. Fig. S7B).

As proposed by the reviewer we replaced strikingly (line 434) by “consistent with this result”.

Lane 335, which positive and negative control did they use for this experiment?

In the revised manuscript, we modified both the text to indicate that VEGF is positive control and the figure legends to better indicate that HPRT is the negative control.

Lane 486 -495 the hear presented data are weak to argue for a joint function and a relevance for GBM.

Regarding the relevance in GBM, we toned down the sentence that was replaced with “*Finally, to investigate the potential clinical importance of our findings,...* ».

We also rephrased the last sentence to highlight that the joint function is strongly supported by a set of evidences.

Could they test if the cells grow slower and migrate less after silencing of hnRNAP?

We believe that the aspects of migration and invasiveness are indeed very interesting to explore. However, in the context of this manuscript that focuses on the DDR, these aspects would appear out of scope and deserve to be fully explored in a future publication.

Reviewer #4

4.1.a.

****I am confused by the above. In table S1, the blue entries are imputed, but the values seem very high. Was the random value imputed as a function of MaxQuant software – why was this method chosen?**

The imputed values are ranging from 19 to 25 roughly but they remain in the lower end of the gaussian distribution of recorded intensities. This may seem high, yet our raw data output gives intensity values starting from roughly 0.5-1.10e6 (around 19 in log2), and no values below this threshold.

In these experiments, we used a function of Perseus, out of Maxquant output data, allowing data imputation with the following explanation from the programmer:

<http://www.coxdocs.org/doku.php?id=perseus:user:activities:matrixprocessing:imputation:replacemissingfromgaussian>

We used the default values: 0.3 as gaussian width relative to the standard deviation of measured values and 1.8 as downshift factor (default values). This information was added to the Material and Method section.

****The description on the front page of table S1 ‘These are the IDs of those proteins that have at least half of the peptides that the leading protein has’ is very confusing – please clarify.**

We agree with the reviewer that the description increases the complexity of the table.

The first column (“Protein IDs”) of the table is the selected protein ID designated as identified. In some cases, other proteins with different IDs may share with the selected protein some of the sequence (only one AA substitution is enough to give a separate ID). In the second column (“Majority proteins ID”), in addition to the protein ID indicated in the first column, the IDs of these other proteins sharing some peptides are mentioned.

To clarify the B column in Suppl. Table S1 we rephrased the description:
‘IDs of proteins sharing at least half the number of peptides with the ID protein in column A’

****Also it would be good on the front page to describe what the blue and red backgrounds refer to on the second page**

As suggested, the description of the Column Q on the front page was modified as to:
“Mean of the Log2-fold change (FC) - Blue and red backgrounds correspond to negative and positive values for the Mean of the Log2-fold change (FC) (Column Q), respectively. ”

4.1.b

****How were these data corrected for multiple testing – this needs to be added to the methods section with a statement about what constitutes a satisfactory q-value.**

The q-Values were obtained by permuted FDR with a limitation at $q < 0.05$. We added this information in the Material and method section.

****This section very much reads as if Perseus was used as a black box by the authors.**

As stated above, in these experiments we used a function of Perseus, out of Maxquant output data, allowing data imputation. More details on data processing were provided in the revised version.

4.1.d

****I am not convinced by the above argument. The authors are saying here that with four replicates any differences between G3AAWT and 7dG are most important and the other controls, although nice to have, are not necessary to determine the difference in these binding partners, but surely the controls are needed to determine background that might be different for the G3AAWT and 7dG interactors.**

We understand the point raised by the reviewer. To provide a quantitative evaluation of the differential protein binding to the sequences forming or not RG4s and to ensure that the identified proteins were not the result of non-specific binding, we performed a quadruplicate analysis and withdrawn from the final list the background proteins binding both the folded and unfolded RG4. To clarify this point, **Suppl. Table S1** now includes the list of these background proteins. We referred to this list in the main text.

We are aware that further studies including several RNAs will be necessary to fully characterize the RG4 proteome in depth. This should involve not only the RNA affinity chromatography-MS but also a throughout characterization of the identified proteins to ensure that the exclusion of certain RBPs from the RG4-binding protein list is really justified. Indeed, as mentioned in previous studies from Kurreck’s lab (Serikawa Biochimie 2018; Von Hacht NAR 2014), the number of protein interactants could be erroneously reduced in a way that depends on the specific control used. This extensive study (which is highly demanding in terms of manpower, time and cost) goes far beyond the scope of this manuscript, which uses the RP-MS analysis as a starting point to then focus on the more representative factors of

the RG4 binding machinery for which we have carried out the requested controls (**Fig. 1 and Suppl. Fig. S2, S3**). Take into account the reviewer's criticisms, we added these considerations in the new version (discussion section).

REVIEWERS' COMMENTS:

Reviewer #2 (Remarks to the Author):

I am very pleased with the made changes and I congratulate the authors on this nice piece of work

Reviewer #4 (Remarks to the Author):

I am satisfied that the newest version of this manuscript addresses all my previous concerns.

I would like to thank the authors for their diligence in responding to reviewers' comments

We would like to thank the 4 reviewers for all the relevant comments and very constructive suggestions that guided us to this new improved version of the manuscript.

REVIEWERS' COMMENTS:

Reviewer #2 (Remarks to the Author):

I am very pleased with the made changes and I congratulate the authors on this nice piece of work

We're delighted that reviewer 2 agreed with our changes.

Reviewer #4 (Remarks to the Author):

I am satisfied that the newest version of this manuscript addresses all my previous concerns. I would like to thank the authors for their diligence in responding to reviewers' comments

We are pleased that reviewer 4 appreciated the efforts we made to clarify the points he raised.